# SPR-RAFT: Parameter-Efficient Regression-Aware Fine-Tuning for Biomedical LLM Regression

Yuanlin Yang [1]  Chenhui Li [2]  Xuhao Guo [3]  Anqi Zhang [4]  Hoi Leong Lee [5]  Haodong Liu [6]

## Abstract

Biomedical regression tasks require predicting continuous targets from heterogeneous and unstructured evidence. While Large Language Models (LLMs) provide a robust interface for reasoning over mixed modalities, they are inherently limited by their discrete tokenization and cross-entropy objectives, which lack awareness of numerical proximity. To bridge this gap, we present **SPR-RAFT**, a parameter-efficient and regression-aware framework that adapts frozen LLMs for high-precision regression. SPR-RAFT introduces a dual-module architecture: a learnable soft prompt that conditions the LLM to route numerical reasoning into a specific latent state, and a lightweight regression head anchored on a dedicated readout token for numerical reasoning consolidation. Crucially, we align these two modalities via a hybrid objective that combines distribution-based text generation with representation-based robust regression. This ensures the model remains both semantically coherent and numerically calibrated. With only about 1.6M trainable parameters ($\sim$0.04% of a 4B-parameter backbone), SPR-RAFT consistently outperforms prompting strategies, standard fine-tuning, and non-LLM baselines across diverse biomedical benchmarks, including clinical trial duration, biological age estimation, and molecular property prediction.

## 1. Introduction

Large language models (LLMs) have emerged as versatile interfaces for reasoning over natural language and semi-structured data. In biomedicine, this capability is uniquely valuable because predictive signals are rarely confined to clean feature vectors. Instead, they mix clinical narratives, trial protocols, eligibility criteria, laboratory panels, units, and missing values. Biomedical pretraining has shown that large transformers can encode substantial biomedical knowledge from corpora such as PubMed and clinical text (Lee et al., 2019; Gu et al., 2021; Luo et al., 2022). This motivates using an LLM as an end-to-end regressor: serialize heterogeneous biomedical evidence into text and predict a single real-valued target.

However, a fundamental mismatch exists between the discrete nature of language modeling and the continuous nature of regression. LLMs generate numbers as token sequences and are typically adapted with token-level cross-entropy, which does not encode proximity between nearby numeric values (Thawani et al., 2021; Zausinger et al., 2024). Decoding strategies further introduces instability, because the most likely string is not necessarily the best prediction under squared error. These issues appear in quantitative evaluations including medical calculation benchmarks (Khandekar et al., 2024), and connect to broader critiques that next-token objectives can be misaligned with downstream metrics (Bachmann & Nagarajan, 2024).

Prior approaches have struggled to simultaneously achieve regression accuracy and parameter efficiency. Recent regression-aware methods, such as RAFT (Lukasik et al., 2025), align the LLM's output distribution with continuous metrics via Bayesian decision theory. However, they typically rely on full-parameter fine-tuning, which is computationally expensive and prone to catastrophic forgetting, especially in low-resource biomedical domains. Conversely, general-purpose Parameter-Efficient Fine-Tuning (PEFT) methods, such as LoRA (Hu et al., 2021), Prefix-Tuning (Li & Liang, 2021), and P-Tuning (Liu et al., 2022), successfully reduce computational costs by freezing the backbone and optimizing low-rank matrices or continuous prompts. Yet, these methods optimize generic next-token likelihood without mechanisms to handle the metric proximity or con-

[1]Department of Electrical Engineering, City University of Hong Kong, Hong Kong SAR, China [2]Computational Science and Engineering, Harvard University, Cambridge, MA, USA [3]School of Software Engineering, South China University of Technology, Guangzhou, Guangdong, China [4]School of Management, Shanghai University of International Business and Economics, Shanghai, China [5]Faculty of Electronic Engineering and Technology, Universiti Malaysia Perlis, Perlis, Malaysia [6]Nanjing University, Nanjing, China. Correspondence to: Anqi Zhang <zhanganqi@suibe.edu.cn>.

*Proceedings of the 43rd International Conference on Machine Learning*, Seoul, South Korea. PMLR 306, 2026. Copyright 2026 by the author(s).

tinuity of numerical targets.

This limitation is especially problematic in biomedicine, where supervision is often scarce and inputs are heterogeneous, featuring pervasive missingness, variable measurement protocols, and inconsistent units (Miotto et al., 2018; Shickel et al., 2018; Xiao et al., 2018; Pathak et al., 2013). Under these conditions, full fine-tuning can be unstable, while zero-shot prompting remains numerically brittle.

To resolve this, we propose **SPR-RAFT** (Soft Prompt Regression-Aware Fine-Tuning), a method designed to align the discrete and continuous modalities within a frozen LLM. Our insight is to decouple the information routing from the value extraction: (i) Soft Prompt: Instead of modifying the massive backbone, we prepend a learnable soft prompt. Its role is to modulate the frozen LLM's attention, effectively guiding the scattered biomedical evidence to be compressed into the hidden state of a special anchor token, `[REG]`. (ii) Regression Head: A lightweight head then reads this compressed representation from `[REG]` to predict a continuous distribution, explicitly modeling uncertainty. By training these components jointly with a robust loss that penalizes both discrete token errors and continuous numerical deviations, SPR-RAFT forces the discrete language model to respect continuous metrics.

We evaluate SPR-RAFT on a diverse suite of biomedical regression tasks, spanning clinical trial forecasting, population health risk modeling, biomedical semantic similarity scoring, and molecular property prediction. Our findings reveal that by modulating a mere ∼0.04% of the parameters in a 4B-parameter backbone, we effectively bridge the discrete-continuous gaptransforming general-purpose LLMs into numerically calibrated regressors that consistently outperform specialized non-LLM basifiers, task-specific encoders, and state-of-the-art LLM adaptation baselines.

Our contributions can be summarized as follows:

(i) We propose SPR-RAFT, the first framework to integrate soft prompt tuning with a specialized regression head anchored on a dedicated `[REG]` readout token. This design fills the gap between computationally heavy regression fine-tuning (e.g., RAFT) and regression-agnostic PEFT methods (e.g., LoRA, P-Tuning), allowing a frozen LLM to perform calibrated regression with only ∼1.6M trainable parameters (∼0.04% of a 4B backbone).

(ii) We propose a dual-branch learning objective that harmonizes the LLM's discrete token generation with continuous regression. By jointly optimizing a robust text-based loss (Huber) and a representation-based uncertainty loss (Gaussian NLL), we resolve the numerical brittleness inherent in standard cross-entropy training.

(iii) We benchmark end-to-end biomedical LLM regression across heterogeneous modalities and show consistent gains.

**Conflict of Interest Disclosure.** The authors declare no financial conflicts of interest. All experiments use publicly released language models (Qwen3, Gemma3, Llama-3.2) and publicly available biomedical benchmarks (TrialBench, NHANES, BIOSSES, MoleculeNet); no author is currently employed by, or holds equity in, the entities producing these models or datasets.

## 2. Related Work

Large transformers pretrained or adapted on biomedical corpora have demonstrated strong biomedical representations and factual priors (Lee et al., 2019; Gu et al., 2021; Luo et al., 2022). These models motivate using LLMs as universal interfaces for biomedical prediction, since they can consume mixed-format biomedical evidence via text serialization rather than task-specific feature engineering.

Representing and predicting numbers in LLMs is challenging because tokenization and string rendering introduce discontinuities and format sensitivity (Thawani et al., 2021; Zausinger et al., 2024). When evaluation lives in continuous space, token-level likelihood can be poorly aligned with regression error. Quantitative evaluations highlight these limitations in medically grounded settings (Khandekar et al., 2024) and more broadly in critiques of next-token objectives for downstream reliability (Bachmann & Nagarajan, 2024).

Regression-aware inference computes continuous predictions from token distributions to better match regression losses without changing model parameters (Lukasik et al., 2024). Regression-aware fine-tuning optimizes a regression-aligned objective during training, shaping probability mass according to continuous error rather than only token identity (Lukasik et al., 2025). TRACT further combines regression-aware optimization with chain-of-thought reasoning for judge-style scoring tasks (Chiang et al., 2025).

PEFT methods adapt LLMs by updating a small number of parameters. LoRA injects low-rank adapters into linear projections and has become a strong default baseline (Hu et al., 2021). Prompt-based approaches learn continuous vectors that condition a frozen model at the input level (Lester et al., 2021) or via layer-wise prefixes (Li & Liang, 2021), and P-Tuning shows that prompt-style adaptation can approach full fine-tuning across scales (Liu et al., 2022).

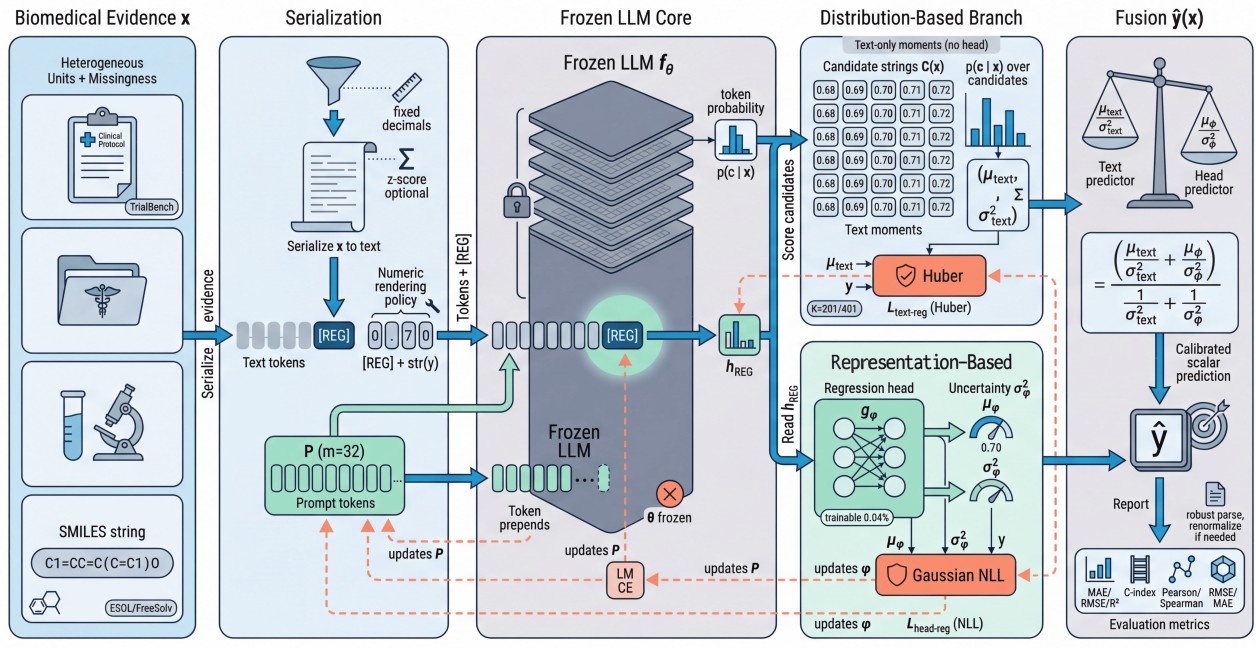

*Figure 1.* **Architecture of SPR-RAFT.** SPR-RAFT adapts a frozen LLM for biomedical regression by decoupling information routing from value extraction. Inputs are serialized into text and the target is formatted as `[REG] + str(y)`. (1) **Information routing:** a trainable soft prompt $P$ steers the frozen backbone $f_\theta$ to compress relevant evidence into the anchor hidden state $h_{\text{REG}}$. (2) **Dual-branch alignment:** a distribution-based branch estimates $(\mu_{\text{text}}, \sigma^2_{\text{text}})$ from candidate-string probabilities (Huber loss), while a representation-based branch maps $h_{\text{REG}}$ through a lightweight head $g_\phi$ (heteroscedastic Gaussian NLL). (3) **Uncertainty-aware fusion:** the final prediction $\hat{y}(x)$ is a precision-weighted combination of the two branches. Dashed arrows indicate that gradients update only $(P, \phi)$ while $\theta$ remains frozen.

## 3. Method

### 3.1. Problem Setup

We study text-to-scalar regression with autoregressive LLMs. Each example consists of an input $x$ (serialized biomedical evidence) and a target $y \in \mathbb{R}$. The model is required to generate an intermediate reasoning rationale (delimited by the special token `[THINK]`), followed by the anchor token `[REG]` and a single canonical numeric string that deterministically maps to $y$ under a fixed rendering policy. `[THINK]` marks a free-form rationale span that is not directly evaluated; `[REG]` designates the numeric answer span and provides a stable hidden-state readout for the regression head. We denote the tokenizer vocabulary by $\mathcal{V}$ and use $p_\theta(\cdot \mid \cdot)$ for the autoregressive distribution induced by the LLM.

A practical challenge is that numbers are generated as discrete token sequences, while evaluation is performed in continuous space. SPR-RAFT addresses this continuous-discrete mismatch by combining (i) a probability-space (distribution-based) estimator that directly aligns token probabilities with regression error and (ii) a representation-space (head-based) estimator that provides dense supervision and calibrated uncertainty.

We serialize heterogeneous evidence into a single text sequence (e.g., protocol fields, lab panels with units, missingness markers). The supervised target is formatted as

$$\text{response} = \text{[THINK]} \ldots \text{[REG]} \, \text{str}(y), \quad (1)$$

where $\text{str}(\cdot)$ is a deterministic numeric renderer (fixed decimals and notation), and `[REG]` is a dedicated anchor token.

We define a deterministic numeric parser $\text{num}(\cdot)$ that isolates and converts the substring strictly following the `[REG]` marker into a real value. The parser returns $\bot$ if the anchor is missing or the suffix is malformed.

Let $f_\theta$ be a pretrained LLM backbone with hidden size $d$ and parameters $\theta$. SPR-RAFT freezes $\theta$ and introduces two trainable components:

**Soft prompt.** A trainable prompt matrix $P \in \mathbb{R}^{m \times d}$ of length $m$ is prepended as continuous "virtual tokens" to the input embedding sequence. Given token embeddings $E(x)$, we form

$$H = f_\theta([P; E(x)]), \quad (2)$$

where $H$ denotes all hidden states. Soft prompts provide a smooth conditioning interface under extremely small parameter budgets. *(This differs from adapter-style updates in*

*that the backbone transformations remain unchanged; only the continuous conditioning is learned.)*

**[REG]-anchored regression head.** Let $h_{\text{REG}} \in \mathbb{R}^d$ be the hidden state at the [REG] position. We apply a lightweight head $g_\phi$ to predict a mean and (optionally) a log-variance:

$$(\mu_\phi(x), s_\phi(x)) = g_\phi(h_{\text{REG}}), \quad \sigma_\phi^2(x) = \exp(s_\phi(x)). \quad (3)$$

We use either a linear head or a small MLP with a bottleneck to keep trainable parameters well below 1% of the backbone.

This decoupling architecture serves a dual purpose: the soft prompt acts as a task-specific lens that focuses the model's vast parametric memory onto the anchor token, while the dual-branch objective ensures that this compressed information is dual-mapped to both the linguistic probability space and the metric embedding space. This synergy effectively regularizes the learning process against the jagged loss landscape of discrete tokenization.

### 3.2. Distribution-Based Regression-Aware Estimation

To explicitly align discrete generation with continuous regression, we compute a regression-aware predictor from token probabilities by scoring a candidate set of numeric strings.

For each task we define a candidate set $\mathscr{C}(x)$ of size $K$ containing numeric strings that follow the same rendering policy as $\text{str}(y)$. For bounded targets, we use a uniform grid in value space. For unbounded or heavy-tailed targets, we use either (i) a clipped uniform grid in a task-specific range, or (ii) a mixed grid that allocates more points near the empirical density peak (details in Appendix B.2). Each candidate $c \in \mathscr{C}(x)$ maps to a value $v(c) = \text{num}(c)$.

Given the context up to [REG], the model assigns each candidate string $c = (c_1, \ldots, c_{|c|})$ an autoregressive likelihood

$$\log \tilde{p}(c \mid x) = \sum_{t=1}^{|c|} \log p_\theta(c_t \mid c_{<t}, x, P), \quad (4)$$

computed by teacher-forced scoring. We then normalize over candidates with a log-sum-exp:

$$p(c \mid x) = \frac{\exp(\log \tilde{p}(c \mid x))}{\sum_{c' \in \mathscr{C}(x)} \exp(\log \tilde{p}(c' \mid x))}. \quad (5)$$

We compute the distributional mean and variance in value space:

$$\begin{aligned} \mu_{\text{text}}(x) &= \sum_{c \in \mathscr{C}(x)} p(c \mid x) v(c), \\ \sigma_{\text{text}}^2(x) &= \sum_{c \in \mathscr{C}(x)} p(c \mid x) \times (v(c) - \mu_{\text{text}}(x))^2. \end{aligned} \quad (6)$$

If $\text{num}(c) = \perp$ for a small subset of candidates, we drop those candidates and renormalize the remaining probability mass.

### 3.3. Regression-Aware Training Objective

SPR-RAFT combines three signals: a regression-aware loss on $\mu_{\text{text}}$, a head-based heteroscedastic regression loss on $\mu_\phi$, and an optional token-level language modeling loss.

We optimize a robust loss on the probability-space estimate:

$$\mathscr{L}_{\text{text}} = \text{Huber}_\delta(\mu_{\text{text}}(x) - y). \quad (7)$$

Huber reduces sensitivity to outliers common in biomedical targets (e.g., heavy-tailed biomarkers) while preserving squared-error behavior near zero.

We train the head with a heteroscedastic Gaussian negative log-likelihood:

$$\mathscr{L}_{\text{head}} = \frac{1}{2} \exp(-s_\phi(x)) (\mu_\phi(x) - y)^2 + \frac{1}{2} s_\phi(x). \quad (8)$$

This yields calibrated uncertainty for fusion and provides dense gradients even when candidate sets are coarse. If variance prediction is disabled, we set $s_\phi(x) = 0$ and recover MSE.

When the output includes a short rationale or structured fields before [REG], we also apply cross-entropy on the output tokens:

$$\mathscr{L}_{\text{LM}} = -\sum_t \log p_\theta(w_t \mid w_{<t}, x, P), \quad (9)$$

optionally restricted to the rationale span and the numeric span (Appendix). For purely numeric outputs, restricting $\mathscr{L}_{\text{LM}}$ to the numeric tokens is sufficient and avoids overfitting stylistic text.

We minimize

$$\mathscr{L} = \lambda_{\text{LM}} \mathscr{L}_{\text{LM}} + \lambda_{\text{text}} \mathscr{L}_{\text{text}} + \lambda_{\text{head}} \mathscr{L}_{\text{head}}, \quad (10)$$

updating only $(P, \phi)$ while keeping $\theta$ frozen. This makes regression-awareness explicit: if $\lambda_{\text{text}} = 0$, the method reduces to head-only PEFT; if the head is removed and only $\mathscr{L}_{\text{text}}$ is used, the method reduces to RAFT-style probability-space regression alignment under PEFT constraints.

### 3.4. Uncertainty-Aware Fusion at Inference

At inference, SPR-RAFT produces two predictors: $(\mu_{\text{text}}, \sigma_{\text{text}}^2)$ from Eq. (6) and $(\mu_\phi, \sigma_\phi^2)$ from Eq. (3). A naive fusion strategy is Inverse Variance Weighting (IVW), which assumes the two estimators are independent. However, this assumption is violated in our setting. Both branches share the same frozen backbone $f_\theta$ and soft

prompt $P$. Consequently, a representational failure in the backbone (e.g., failing to capture a negation in the input text) propagates to both the token distribution and the regression head, inducing a positive correlation between their errors.

Directly applying IVW under positive correlation leads to variance underestimation, causing the model to be overconfident. To mitigate this, we adopt a Generalized Least Squares (GLS) fusion strategy that accounts for the correlation coefficient $\rho$ between the estimators. The fused prediction $\hat{y}(x)$ is computed as:

$$\hat{y}(x) = \frac{(\sigma_\phi^2 - \rho\sigma_{\text{text}}\sigma_\phi)\mu_{\text{text}} + (\sigma_{\text{text}}^2 - \rho\sigma_{\text{text}}\sigma_\phi)\mu_\phi}{\sigma_{\text{text}}^2 + \sigma_\phi^2 - 2\rho\sigma_{\text{text}}\sigma_\phi}, \quad (11)$$

where dependencies on $x$ for $\mu$ and $\sigma$ are omitted for brevity. This formulation reduces the weight of the estimator that is more correlated with the other, preventing information redundancy. The resulting calibrated variance is:

$$\sigma_{\text{fused}}^2 = \frac{\sigma_{\text{text}}^2 \sigma_\phi^2 (1 - \rho^2)}{\sigma_{\text{text}}^2 + \sigma_\phi^2 - 2\rho\sigma_{\text{text}}\sigma_\phi}. \quad (12)$$

When $\rho = 0$, this recovers the standard IVW solution; as $\rho \to 1$, the fusion becomes conservative, preventing the variance collapse observed in naive IVW. In practice, we treat $\rho$ as a learnable scalar parameterized via a sigmoid, $\rho = \sigma(w_\rho)$ with $w_\rho \in \mathbb{R}$, which constrains $\rho \in (0, 1)$. Across our benchmarks, $\rho$ converges to the range $[0.65, 0.85]$, confirming that the two branches share substantial residual error structure.

### 3.5. Efficiency and Parameter Budget

The soft prompt contributes $md$ trainable parameters. The regression head consists of a two-layer MLP with a bottleneck of width $d/4$ and a final scalar projection, contributing $d \times d/4 + d/4 \times 1$ parameters. In our default configuration with the Qwen3-4B backbone ($d = 2560$, $m = 32$), the trainable budget is $32 \times 2560 + 2560 \times 640 + 640 \approx 1.72\text{M}$ parameters, or roughly 0.04% of the 4B-parameter backbone. With variance prediction enabled, the head emits two outputs and the budget grows by a single $640 \times 1$ projection (negligible).

Computationally, distribution-based estimation requires scoring $K$ candidates. With average candidate length $L$, the additional cost is $O(KL)$ token likelihood evaluations, which can be batched efficiently. We use $K = 32$ for all main results (sensitivity to $K \in \{16, 32, 64\}$ in Appendix B.1) and keep the numeric string short via a fixed rendering policy.

## 4. Experiments

### 4.1. Experimental Setup

We evaluate SPR-RAFT as an end-to-end *LLM regressor* on biomedical regression benchmarks. Across tasks, each example is a serialized input string $x$ (text or textified structured data) paired with a real-valued target $y \in \mathbb{R}$. Unless stated otherwise, we use `Qwen3-4B` as the backbone (hidden size $d = 2560$).

To reduce brittleness from numeric tokenization and surface-form variance, we (i) enforce a fixed numeric rendering policy (fixed-point or scientific notation with a fixed number of decimals) and (ii) apply a robust numeric parser that strips punctuation, whitespace, and common units when applicable. Per-dataset rendering policies are reported in Appendix B.3.

We train with AdamW, learning rate $5 \times 10^{-4}$ for soft prompts and regression head, weight decay in $[0, 0.01]$, and an effective batch size of 16 (per-GPU batch 4 with 4 GPUs). Unless stated otherwise, we use a soft prompt length $m = 32$, a candidate set size $K = 32$, $\lambda_{\text{LM}} = \lambda_{\text{text}} = \lambda_{\text{head}} = 1.0$, Huber threshold $\delta = 1.0$, and train for 5 epochs with early stopping on validation performance (patience 5 epochs). We repeat each experiment with 3 random seeds and report the mean $\pm$ standard deviation. Full hyperparameter details are provided in Appendix B.1, and perdataset prompt templates are provided in Appendix B.4.

### 4.2. Benchmarks

We evaluate SPR-RAFT on biomedical regression benchmarks covering clinical trials (TrialBench) (Chen et al., 2024), population health (NHANES-based biological age with linked mortality evaluation (National Center for Health Statistics, 2019)), biomedical semantic similarity (BIOSSES) (Sogancioglu et al., 2017), and molecular property prediction (ESOL and FreeSolv from MoleculeNet) (Wu et al., 2018; Delaney, 2004; Mobley & Guthrie, 2014). We compare against strong non-LLM regressors, discriminative biomedical encoders, and LLM baselines. Additional benchmark descriptions and implementation details are provided in Appendix A.

### 4.3. Baselines

We compare SPR-RAFT against both LLM-based and non-LLM baselines.

**Non-LLM baselines.** For tabular NHANES, we include standard strong baselines: Elastic Net, XGBoost, MLP regressor, and transformer-based tabular models (e.g., FT-Transformer or TabTransformer), all trained on identical splits and preprocessing.

*Table 1.* TrialBench regression results. We report the most challenging phase for each task (Phase III for duration; Phase IV for dropout). Multimodal DL results are taken from Chen et al. (2024) when available. ↓ indicates lower is better and ↑ indicates higher is better.

| Method | Duration (years) | | | | Dropout rate (0–1) | | | |
|---|---|---|---|---|---|---|---|---|
| | MAE ↓ | RMSE ↓ | $R^2$ ↑ | Phase | MAE ↓ | RMSE ↓ | $R^2$ ↑ | Phase |
| Multimodal DL | 0.83 | 1.26 | 0.65 | I | 0.45 | 0.46 | 0.63 | I |
| XGBoost | 0.88 | 1.32 | 0.62 | I | 0.48 | 0.55 | 0.61 | I |
| Zero-shot LLM | 3.60 ±0.21 | 8.28 ±0.43 | -0.18 | I | 0.82 ±0.04 | 0.84 ±0.05 | 0.01 | I |
| CE Fine-tune (LLM) | 1.78 ±0.12 | 2.20 ±0.18 | 0.27 | I | 0.63 ±0.05 | 0.68 ±0.06 | 0.35 | I |
| RAFT (LLM) | 0.94 ±0.06 | 1.34 ±0.09 | 0.60 | I | 0.41 ±0.03 | 0.43 ±0.03 | 0.68 | I |
| **SPR-RAFT (ours)** | **0.70** ±0.03 | **1.08** ±0.06 | **0.73** | I | **0.39** ±0.02 | **0.42** ±0.03 | **0.70** | I |
| Multimodal DL | 1.44 | 1.84 | 0.32 | III | 0.42 | 0.44 | 0.22 | IV |
| XGBoost | 1.55 | 1.85 | 0.30 | III | 0.43 | 0.49 | 0.19 | IV |
| CE Fine-tune (LLM) | 6.30 ±0.42 | 9.70 ±0.65 | 0.09 | III | 0.81 ±0.05 | 0.93 ±0.06 | 0.13 | IV |
| RAFT (LLM) | 1.63 ±0.10 | 2.38 ±0.15 | 0.24 | III | 0.42 ±0.03 | 0.44 ±0.03 | 0.21 | IV |
| **SPR-RAFT (ours)** | **1.15** ±0.07 | **1.55** ±0.10 | **0.39** | III | **0.36** ±0.02 | **0.41** ±0.03 | **0.34** | IV |

*Table 2.* Performance on the NHANES dataset (2007–2018) with mortality follow-up. ↑ indicates higher is better.

| Method | AUC ↑ | Mortality C-index ↑ |
|---|---|---|
| Elastic Net | 0.77 | 0.76 |
| **XGBoost** | **0.87** | **0.86** |
| FT-Transformer | 0.86 | 0.84 |
| Zero-shot LLM | 0.36 ±0.04 | 0.35 ±0.04 |
| LoRA Fine-tune | 0.60 ±0.03 | 0.60 ±0.03 |
| CE Fine-tune | 0.59 ±0.04 | 0.60 ±0.04 |
| RAFT | 0.75 ±0.02 | 0.75 ±0.02 |
| SPR-RAFT | 0.84 ±0.02 | 0.84 ±0.02 |

*Table 3.* BIOSSES regression results. Results for BERT-based are taken from Sogancioglu et al. (2017). ↑ indicates higher is better.

| Method | Pearson $r$ ↑ | Spearman $\rho$ ↑ |
|---|---|---|
| BERT-based | 0.92 | 0.91 |
| Zero-shot LLM | 0.56 ±0.03 | 0.66 ±0.04 |
| LoRA Fine-tune | 0.66 ±0.03 | 0.75 ±0.03 |
| CE Fine-tune | 0.68 ±0.04 | 0.74 ±0.04 |
| RAFT | 0.81 ±0.02 | 0.89 ±0.02 |
| **SPR-RAFT** | **0.93** ±0.01 | **0.95** ±0.01 |

**LLM baselines.** We compare against: (i) **Zero-shot prompting**: instruction-only numeric prediction with fixed formatting, (ii) **LoRA fine-tuning**: parameter-efficient fine-tuning with low-rank adapters (Hu et al., 2021), (iii) **Supervised CE fine-tuning**: fine-tune on numeric strings with cross-entropy only, (iv) **RAFT**: regression-aware fine-tuning without soft prompts (Lukasik et al., 2025), (v) **Prompt tuning baselines**: soft prompt tuning with CE-only loss, and prefix-style tuning where applicable (Liu et al., 2022). For all LLM methods, we standardize input serialization, maximum context length, numeric rendering, decoding strategy, and the candidate set construction for regression-aware estimation.

### 4.4. Main Results

#### 4.4.1. TRIALBENCH: TRIAL DURATION AND DROPOUT

As summarized in Table 1, SPR-RAFT outperforms all baseline models across all benchmarks. Specifically, on the TrialBench Duration task, our method achieves a 25.5%

relative reduction in MAE compared to the strongest LLM baseline (RAFT). This improvement is statistically significant with $p < 0.01$ based on a two-tailed t-test over five independent runs with different random seeds.

#### 4.4.2. NHANES: BIOLOGICAL AGE AND MORTALITY CONCORDANCE

Table 2 reports results on NHANES (2007–2018) with mortality follow-up. Unlike TrialBench, NHANES is a large-scale purely tabular benchmark where predictive signal is dominated by structured covariates, monotone effects, and local feature interactions. This setting strongly favors tabular specialists such as XGBoost, which achieves the best AUC and C-index, and leaves limited room for LLM-based regressors whose main advantage is handling heterogeneous and unstructured evidence.

Despite this mismatch, SPR-RAFT still consistently improves over standard LoRA (+24%) and RAFT (+9%), reaching competitive performance. These results indicate that regression-aware alignment remains effective even when inputs are textified tables, and clarify that NHANES

*Table 4.* Molecular property regression RMSE (lower is better). Comparison with graph-based models, zero-shot prompting, and representative LLM baselines.

| Method | ESOL | FreeSolv | Lipophilicity |
|---|---|---|---|
| *Graph Neural Networks* | | | |
| GIN (Xu et al., 2019) | 1.243 | 2.871 | 0.781 |
| Graphormer (Ying et al., 2021) | 0.901 | 2.210 | 0.740 |
| *LLM-based Models* | | | |
| GIMLET (Zero-Shot) (Zhao et al., 2023) | 1.132 | 5.103 | 1.345 |
| Zero-shot LLM | 1.050 | 7.624 | 4.635 |
| LoRA Fine-tune | 0.968 | 7.546 | 2.241 |
| CE Fine-tune | 0.704 | 1.353 | 1.745 |
| *Recent LLM Baselines* | | | |
| Mol-LLM (Generalist) (Lee et al., 2025) | 0.607 | 0.874 | 0.858 |
| LlaSMol (Generalist) (Yu et al., 2024) | 0.629 | 0.883 | 0.871 |
| InstructMol (Specialist) (Cao et al., 2024) | **0.407** | 0.941 | 0.753 |
| BioT5+ (Pei et al., 2024) | 0.642 | 0.969 | 0.897 |
| *Regression-Aware Fine-Tuning* | | | |
| RAFT | 0.850 ±0.04 | 1.881 ±0.09 | 0.945 ±0.04 |
| **SPR-RAFT (ours)** | 0.597 ±0.03 | **0.455** ±0.04 | **0.664** ±0.03 |

is a regime where dedicated tabular models can remain superior while SPR-RAFT provides a unified, parameter-efficient LLM regressor.

### 4.4.3. BIOMEDICAL STS REGRESSION

Table 3 reports results on BIOSSES. SPR-RAFT reaches a Spearman $\rho$ of 0.95, outperforming the specialized RAFT baseline by 6.7% and matching state-of-the-art discriminative BERT models. The improvement over CE-only fine-tuning is particularly striking (+21% in Pearson $r$, $p < 0.005$), suggesting that the dual-branch objective effectively harmonizes the models linguistic understanding of biomedical nuances with the continuous scale of semantic similarity.

### 4.4.4. MOLECULAR PROPERTY PREDICTION

As shown in Table 4, SPR-RAFT achieves state-of-the-art performance on several MoleculeNet benchmarks, notably outperforming specialized multimodal molecular LLMs such as Mol-LLM and LlaSMol. In the FreeSolv hydration free energy task, SPR-RAFT yields an RMSE of 0.455, a relative reduction of 47.9% over the strongest LLM baseline ($p < 0.01$). On the Lipophilicity dataset, we observe a similar trend with an RMSE of 0.664, surpassing graph-based specialists like Graphormer. This superior performance on out-of-distribution chemical spaces demonstrates that by aligning the LLMs representation with physical-chemical gradients, SPR-RAFT successfully transitions from simple SMILES pattern matching to a more profound understanding of molecular property continuity.

### 4.5. Latent Manifold Alignment Analysis

SPR-RAFT substantially outperforms probability-space regression alignment (RAFT) under a comparable parameter budget. To probe the underlying difference, we analyze the geometry of the anchor representation used by our head. For each test example, we extract the final-layer hidden state at the [REG] position, denoted as $h_{\text{REG}} \in \mathbb{R}^d$. We then fit PCA on $\{h_{\text{REG}}\}$ and visualize the first two principal components, coloring points by the (normalized) ground-truth target $y$. As summary statistics, we report the Spearman rank correlation between PCA-1 and $y$, and the coefficient of determination from a linear probe $y \sim$ PCA-1 (PCA sign is arbitrary).

Figure 2 shows a clear progression. In the zero-shot regime, $h_{\text{REG}}$ forms a compact, roughly spherical cloud with no consistent color gradient, indicating that the pretrained backbone does not explicitly encode numeric magnitude along a dedicated direction at the anchor token. RAFT induces a visible ordering along a dominant axis, consistent with probability-space regression losses reshaping token likelihoods to better respect continuous error. However, the embedding cloud remains diffuse: substantial variability persists orthogonal to the regression direction, suggesting that $h_{\text{REG}}$ still entangles magnitude with input-dependent factors such as surface form, context composition, and residual reasoning patterns. In contrast, SPR-RAFT yields a substantially cleaner geometry in which PCA-1 becomes tightly coupled with $y$ (high Spearman and high $R^2$), while the remaining variation is pushed into orthogonal directions with weak association to $y$. Overall, this diagnostic sup-

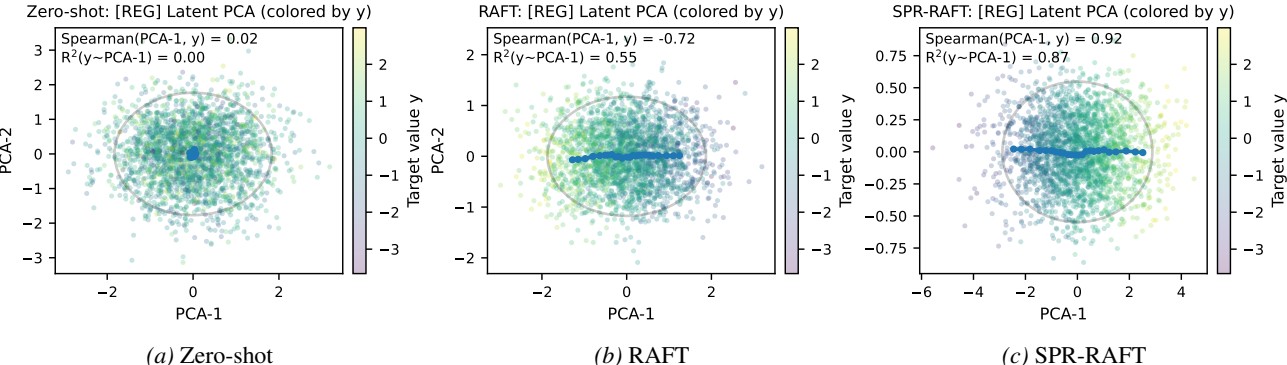

*(a)* Zero-shot        *(b)* RAFT        *(c)* SPR-RAFT

*Figure 2.* **Latent Manifold Alignment of the [REG] Anchor.** We project the hidden states $h_{REG}$ onto the first two principal components (PCs) across test examples, with points colored by the z-scored ground-truth target $y$. (a) Zero-shot: The pretrained backbone exhibits a stochastic, spherical distribution with no inherent numerical ordering. (b) RAFT: Probability-space alignment induces a visible gradient along a dominant axis, yet suffers from high variance due to entanglement with surface forms. (c) SPR-RAFT: Our dual-branch objective yields a clean, monotonic manifold where PCA-1 is tightly coupled with $y$ (Spearman $\rho = 0.92, R^2 = 0.87$), demonstrating effective routing of evidence into a low-dimensional, numerically calibrated representation.

*Table 5.* Ablation study on TrialBench Duration (Phase I). Unless otherwise stated, the backbone is Qwen3-4B and $m = 32$.

| Method Variant | MAE $\downarrow$ | $\Delta$ |
|---|---|---|
| *1. Ablations* | | |
| SPR-RAFT (Dual Branch Fusion, GLS) | **0.70** ±0.03 | - |
|    w/ Naive IVW fusion ($\rho{=}0$) | 0.73 ±0.04 | +0.03 |
| Soft Prompt Only | 0.78 ±0.05 | +0.08 |
| Head Only | 0.74 ±0.04 | +0.04 |
| *2. Impact of Soft Prompt Length (m)* | | |
| SPR-RAFT ($m = 16$) | 0.72 ±0.04 | +0.02 |
| SPR-RAFT ($m = 32$, Default) | 0.70 ±0.03 | - |
| SPR-RAFT ($m = 64$) | 0.69 ±0.03 | -0.01 |
| *3. Variations on Base LLMs* | | |
| Llama-3.2-3B | 0.93 ±0.05 | +0.23 |
| Gemma3-4B | 0.80 ±0.04 | +0.10 |
| Qwen3-8B | 0.68 ±0.03 | -0.02 |

ports our design intent: the soft prompt routes task-relevant evidence into the anchor state, and the lightweight head imposes an explicit representational constraint that promotes a low-dimensional, approximately linear readout for continuous targets, improving both accuracy and stability.

### 4.6. Ablation Studies

We ablate key design choices to isolate the sources of improvement, using TrialBench Duration (Phase I) as a representative benchmark.

Specifically, we compare (i) Soft Prompt Only (no regression head; distribution-based estimation only) and (ii) Head Only (no soft prompt; frozen backbone with a regression head).

To examine the trade-off between capacity and efficiency,

we vary the soft-prompt length $m \in \{16, 32, 64\}$ and report performance as a function of the number of trainable parameters. We further compare a linear regression head with a lightweight MLP head.

To assess the effect of model scale and architecture, we additionally evaluate Gemma3-4B, Qwen3-8B, and Llama-3.2-3B as alternative frozen backbones, holding all other hyperparameters fixed. Across families, SPR-RAFT consistently outperforms the corresponding RAFT baseline (Llama-3.2-3B: $1.92 \to 0.93$; Gemma3-4B: $1.05 \to 0.80$; Qwen3-8B: $0.88 \to 0.68$), demonstrating promising transfer across model families and scales rather than dependence on a specific backbone.

To study the impact of numeric rendering, we test fixed-point versus scientific notation and vary the number of decimal places in $\{1, 2, 3\}$. We further measure boundary sensitivity by evaluating errors on extreme quantiles of the target distribution.

We provide an additional diagnostic plot for the uncertainty-aware fusion in Appendix D (Figure 3), which shows that fusion is most helpful in the head's high-uncertainty region and reduces MSE from 1.345 (text-only) to 1.129 (fusion).

## 5. Limitations

SPR-RAFT improves regression with a strict parameter budget, but several limitations remain.

The distribution-based branch depends on a predefined candidate set $\mathscr{C}(x)$ and a fixed numeric rendering policy. If the candidate range is misspecified, the grid is too coarse, or the target distribution is heavy-tailed, the estimated moments $(\mu_{\text{text}}, \sigma_{\text{text}}^2)$ can be biased and may under-represent tail risk. Regression-aware scoring requires evaluating

$K = 32$ candidates at inference, which adds latency relative to single-pass decoding. While batching and short numeric strings keep this overhead modest in practice, this cost can matter for long contexts or high-throughput settings; head-only inference is cheaper but may lose calibration and robustness. We discuss the empirical latency–accuracy trade-off in Appendix B.1. The method assumes targets can be faithfully represented as canonical numeric strings and recovered by a deterministic parser. Real biomedical pipelines introduce ambiguity from rounding, unit conventions, and measurement protocols, which must be handled in preprocessing and serialization. Candidate-based variance reflects uncertainty within $\mathscr{C}(x)$, and head variance is learned under a Gaussian likelihood. Both can be miscalibrated under distribution shift. Finally, our evaluation covers representative biomedical benchmarks, but broader validation across more clinical datasets, backbone families, and shift scenarios (including rare subpopulations and missingness shifts) is needed. SPR-RAFT is a modeling technique and is not a clinical decision system; any high-stakes use would require external validation, expert oversight, and appropriate governance.

## 6. Conclusion

We presented **SPR-RAFT**, a parameter-efficient and regression-aware adaptation framework that turns a frozen autoregressive LLM into a calibrated text-to-scalar regressor. SPR-RAFT decouples information routing from value extraction by combining a learnable soft prompt with a lightweight `[REG]`-anchored regression head. Training jointly aligns probability-space behavior with continuous error and imposes representation-space supervision with heteroscedastic uncertainty, and inference fuses both predictors using correlation-aware GLS to avoid overconfident uncertainty.

Across biomedical regression tasks including clinical trial forecasting, population health risk modeling, biomedical semantic similarity scoring, and molecular property prediction, SPR-RAFT improves accuracy and calibration over prompting, CE-only tuning, and regression-aware baselines while updating only ~1.6M parameters (~0.04% of a 4B backbone). Because the trainable footprint is a single hot-swappable adapter of about $1.6\,\mathrm{MB}$, the same backbone can be specialized to many regression tasks without duplicating model weights, enabling lightweight deployment in resource-constrained biomedical settings. More broadly, the approach suggests a practical recipe for continuous–discrete alignment in autoregressive models. Promising directions include adaptive candidate proposals with strict train-only statistics, faster regression-aware estimation (for example coarse-to-fine or distillation), stronger uncertainty calibration under distribution shift, and extension to mul-

timodal LLMs where heterogeneous evidence (text, structured records, images, molecular graphs) must be jointly mapped to continuous targets.

## Acknowledgements

We thank the anonymous ICML 2026 reviewers and the area chair for their constructive feedback that substantially improved the manuscript.

## Impact Statement

This work aims to improve the reliability of real-valued prediction with autoregressive language models under parameter-efficient adaptation. The primary intended impact is positive: better-aligned regression objectives and calibrated uncertainty can reduce numerical brittleness in scientific and biomedical prediction settings, where small mistakes may propagate into downstream analyses. By updating only a small number of parameters, SPR-RAFT can also lower the computational and storage cost of adapting models to specialized regression tasks, which may broaden access for research groups with limited resources.

There are, however, important risks. First, biomedical regression outputs can be misinterpreted as clinical guidance. Even when trained on biomedical datasets, SPR-RAFT does not provide medical advice, does not establish causality, and may fail under distribution shift, missingness patterns, or unmodeled confounders. Second, uncertainty estimates may be miscalibrated in out-of-distribution scenarios, which could lead to overconfidence if used without appropriate safeguards. Third, using patient-level or sensitive health data raises privacy and governance concerns, including the possibility of memorization or leakage if models are trained on improperly handled data.

We mitigate these concerns in the following ways. Our method is evaluated on research benchmarks and is presented as a modeling technique rather than a clinical decision system. We recommend that any real-world deployment be paired with domain expert oversight, dataset governance, and rigorous external validation on representative populations. For privacy, the approach is compatible with data-minimizing practices because it freezes the backbone and learns a small set of task parameters, but this does not by itself guarantee privacy; practitioners should still follow established de-identification, access control, and auditing protocols. Finally, we encourage reporting of failure cases, subgroup analyses, and calibration diagnostics to avoid overstating reliability.

Overall, SPR-RAFT is best viewed as a tool for improving numerical faithfulness and calibration in text-to-scalar modeling, with benefits for scientific workflows when used

responsibly and with appropriate validation.

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

# A. Benchmarks

## A.1. Clinical Trial Regression: TrialBench

TrialBench provides multiple prediction tasks derived from clinical trial records. We focus on two regression tasks directly aligned with numeric forecasting: (i) Trial Duration Forecasting (years; 141,940 instances), and (ii) Patient Dropout Rate Forecasting (fraction in $[0, 1]$; 62,058 instances). We evaluate on phase-specific subsets (Phase I to IV) using the official splits.

We serialize each trial into a structured textual prompt containing title, conditions, interventions, phase, enrollment, eligibility criteria, and other key metadata fields. We apply light normalization (canonical field ordering, truncation to 2,048 tokens, and minimal text cleanup).

### A.1.1. CLINICAL TRIAL DURATION PREDICTION

The first regression task focuses on predicting the duration of a clinical trial. The objective is to estimate the continuous trial duration (in years) from initiation to completion, formulated as a single-target regression problem.

The input features for this task are multi-modal, integrating five types of trial information: molecular structure of investigational drugs (e.g., SMILES representations), disease information encoded by ICD-10, free-text descriptions (such as summaries and eligibility criteria), structured numeric and categorical attributes (e.g., trial phase, enrollment size), and MeSH-based semantic descriptors. In total, the dataset provides 35 features per trial instance for the duration prediction task.

For label construction, only trials with both a recorded start date and an actual completion date are retained, while records that only contain an expected completion time are excluded. Furthermore, trials with extremely long durations (greater than 10 years) are removed to mitigate the impact of outliers and support stable regression training. After filtering, the resulting dataset contains approximately 141,940 clinical trials for training and evaluating duration prediction models.

**Illustrative Example.** Consider a phase II oncology trial investigating a small-molecule inhibitor for lung cancer. The corresponding TrialBench instance includes the SMILES string of the investigational compound, ICD-10 codes for lung cancer, detailed eligibility criteria in free-text form (e.g., performance status, prior therapies), and structured fields such as "phase II", anticipated enrollment of 150 participants, and multi-center design. Given these multi-modal features, the duration prediction model outputs a continuous estimate, for example 2.4 years, representing the expected time from trial initiation to completion.

### A.1.2. PATIENT DROPOUT RATE PREDICTION

The second regression task centers on predicting the patient dropout rate. This is formulated as part of a dual-objective learning problem, consisting of: (i) a binary classification task indicating whether patient dropout occurs, and (ii) a regression task predicting the dropout rate, defined as the proportion of enrolled participants who discontinue the trial. The dropout rate is represented as a continuous value in the interval $[0, 1]$.

To construct the regression labels, the dataset extracts, for each eligible trial, the number of participants who dropped out and the total number of enrolled participants from ClinicalTrials.gov. The dropout rate is then computed as the ratio of dropouts to total enrollment, providing a trial-level estimate of overall attrition.

The input features for dropout prediction also combine all five modalities of protocol-level information, similar to the duration task. This includes textual eligibility criteria, target disease descriptors, drug-related features, and trial design characteristics, resulting in 38 features per instance for the dropout prediction dataset. Only trials with explicitly reported dropout counts and enrollment numbers are kept, yielding approximately 62,058 trials used jointly for training the classification and regression components of the dropout prediction task.

**Illustrative Example.** As an example, consider a phase III cardiovascular trial enrolling 1,000 patients. The TrialBench representation includes the drug's molecular structure, ICD-10 codes for the targeted cardiovascular condition, multi-center randomized controlled design, and detailed inclusion/exclusion criteria describing comorbidities and concomitant medications. Based on these multi-modal protocol features, the dropout prediction model estimates a continuous dropout rate, for instance 0.18, which corresponds to a predicted 18% of participants discontinuing the trial before completion.

### A.2. Population Health Regression: NHANES Biological Age

We construct a population-level dataset by merging NHANES 2007–2018 cycles with mortality follow-up and selected measurement panels (BIOPRO, BMX, BPX, CBC, CRP/HSCRP, DEMO, MORT)[1].

We evaluate performance on two complementary objectives: (i) Biological Age Estimation (continuous; years), formulated as a supervised regression task; and (ii) Mortality Risk Stratification, where the predicted biological age serves as a risk proxy, evaluated via the concordance index (C-index) and AUC.

Each individual record is converted to a compact, human-readable summary: demographics + a list of measurements with units where available (e.g., blood pressure, BMI, CRP, CBC panel). Missing values are explicitly marked as NA. We optionally apply log transform to skewed biomarkers (e.g., CRP) and z-score normalization per feature group, but the model ultimately receives the serialized values.

### A.3. Biomedical Semantic Similarity Regression: BIOSSES

We evaluate on BIOSSES, a biomedical sentence-pair semantic similarity benchmark with gold similarity scores. Given two biomedical sentences $s_1, s_2$, the model predicts a similarity score on the dataset scale. We report Pearson's $r$ and Spearman's $\rho$.

We prompt the model with $(s_1, s_2)$ and an instruction to output a single numeric score with a fixed format (one decimal place).

### A.4. Molecular Property Regression from SMILES: ESOL, FreeSolv, Lipophilicity

Accurate prediction of physicochemical properties such as solubility and lipophilicity is central to early-stage drug discovery. We evaluate on three standard MoleculeNet regression datasets where the input is a SMILES string and the target is a real-valued property: ESOL (LogS, including 1,128 molecules), FreeSolv (hydration free energy, including 642 molecules), and Lipophilicity (LogD, including 4,200 molecules). We report RMSE. We use official splits when provided; otherwise we adopt scaffold splits and report mean $\pm$ standard deviation over 3 seeds.

We present the SMILES string with minimal context (property name, units when known, and an expected value range when available) and instruct the model to output a single numeric value in the enforced rendering format.

## B. Additional Experimental Details

### B.1. Training Hyperparameters

Unless stated otherwise, we follow the default configuration used in Section 4. We train only the soft prompt parameters $P$ and the regression head parameters $\phi$.

- **Backbone:** Qwen3-4B.

- **Soft prompt length:** $m = 32$ (ablated in Table 5).

- **Optimizer:** AdamW.

- **Learning rate:** $5 \times 10^{-4}$ (applied to both $P$ and $\phi$).

- **Weight decay:** in $[0, 0.01]$.

- **Effective batch size:** 16.

- **Training length:** 5 epochs with early stopping on validation performance (patience 5 epochs).

- **Candidate set size for distribution-based estimation:** $K = 32$ by default (see Appendix B.2).

---

[1]All data is publicly available at the National Center for Health Statistics website: https://www.cdc.gov/nchs/nhanes/index.html

## B.2. Candidate Set Construction for Regression-Aware Estimation

We use a candidate set size of $K = 32$ unless stated otherwise. We construct candidates using a fixed numeric grid (uniform for bounded targets; density-aware mixed grids for unbounded/heavy-tailed targets) and compute $\hat{y}(x)$ as the probability-weighted expectation under the enforced numeric rendering.

In Section 3.2, we introduced a candidate set $\mathscr{C}(x)$ to discretize the regression target space. Here, we detail the construction strategies for different target distributions.

**Bounded Targets.** For tasks with strictly bounded targets $[y_{\min}, y_{\max}]$ (e.g., semantic similarity scores in $[0, 4]$ or dropout rates in $[0, 1]$), we employ a simple uniform grid:

$$v_k = y_{\min} + \frac{k-1}{K-1}(y_{\max} - y_{\min}), \quad k = 1, \ldots, K. \tag{13}$$

This ensures equidistant coverage across the entire feasible range.

**Unbounded or Heavy-Tailed Targets (Density-Aware Mixed Grid).** For tasks with unbounded or heavy-tailed distributions (e.g., biological age, trial duration), using a uniform grid over a wide range implies poor resolution in high-density regions. To address this, we implement a **Quantile-Based Mixed Grid** strategy that allocates candidates proportional to the empirical data density.

Let $\mathscr{D}_{\text{train}} = \{y_i\}_{i=1}^N$ be the set of target values in the training split. We construct the candidate set of size $K$ via the following steps:

1. **Effective Range Definition:** We first determine the effective support $[V_{\min}, V_{\max}]$ to mitigate the impact of extreme outliers. We calculate the empirical quantiles $q_{\text{low}}$ (0.5%) and $q_{\text{high}}$ (99.5%) of $\mathscr{D}_{\text{train}}$ and set $V_{\min} = q_{\text{low}}$ and $V_{\max} = q_{\text{high}}$.

2. **Anchor Point Selection:** We select $M$ anchor points $\{a_1, \ldots, a_M\}$ (where $M \ll K$) to define the skeleton of the grid. These anchors are chosen based on linearly spaced cumulative probabilities, ensuring they adapt to the distribution shape. Let $Q(\cdot)$ be the empirical quantile function of $\mathscr{D}_{\text{train}}$. We define:

$$a_j = Q\left(p_{\text{low}} + \frac{j-1}{M-1}(p_{\text{high}} - p_{\text{low}})\right), \quad j = 1, \ldots, M, \tag{14}$$

where $p_{\text{low}}$ and $p_{\text{high}}$ are the cumulative probabilities corresponding to $V_{\min}$ and $V_{\max}$ (i.e., 0.005 and 0.995). This naturally places dense anchors near the distribution peak and sparse anchors in the tails.

3. **Piecewise Linear Filling:** We populate the full candidate set by performing linear interpolation between adjacent anchors. We allocate an equal budget of candidates, $N_{\text{fill}} \approx \frac{K}{M-1}$, to each interval $[a_j, a_{j+1}]$.

This approach ensures that the resolution of the probability-space estimator matches the information density of the training data. Finally, all candidate values $v_k$ are rendered into strings $c_k = \text{str}(v_k)$ using the fixed task-specific rendering policy.

## B.3. Numeric Rendering Policies

We enforce per-dataset numeric formats: TrialBench: fixed-point with 2 decimals; NHANES: fixed-point with 2 decimal; BIOSSES: fixed-point with 1 decimal on [0,4]; MoleculeNet: fixed-point with 3 decimals / scientific notation for wide ranges. We also list unit handling and parsing rules (e.g., stripping "mg/dL").

## B.4. Input Templates

We provide the exact serialization templates for TrialBench trials, NHANES records, BIOSSES sentence pairs, and SMILES prompts. Below we include two concrete examples for clarity.

**Example: BIOSSES (sentence-pair similarity).**

```
[INST]
Given two biomedical sentences, output a similarity score in [0,4] with one decimal.
Sentence 1: <s1>
Sentence 2: <s2>
Answer format: [THINK] <brief reasoning> [REG] <score>
[/INST]

a) [THINK] The two sentences describe the same mechanism with minor wording.
   [REG] 3.6
```

**Example: NHANES (biological age).**

```
[INST]
Estimate biological age (years) from the following record.
Output format: [THINK] <rationale> [REG] <age>
[/INST]

Sex: Female
Age: 52
BMI: 28.4 kg/m^2
CRP: 3.10 mg/L
Systolic BP: 138 mmHg
...

[THINK] Elevated inflammation and BP suggest higher biological age.
[REG] 58.2
```

### B.5. Compute, Runtime, and Parameter Counts

We report: (i) trainable parameter counts for each prompt length, (ii) wall-clock training time per epoch, (iii) inference overhead from regression-aware estimation, and (iv) GPU type and memory used in all experiments.

With the default Qwen3-4B backbone ($d = 2560$), the trainable parameter budget decomposes as: soft prompt of length $m = 32$ contributes $32 \times 2560 = 81{,}920$ parameters; the regression head consists of a two-layer MLP $d \to d/4 \to 1$ contributing $2560 \times 640 + 640 = 1{,}639{,}040$ parameters. With heteroscedastic variance prediction enabled, an additional $640 \times 1$ projection adds 640 parameters. The total trainable footprint is approximately 1.72M parameters, or $\approx 0.04\%$ of the 4B-parameter backbone. Stored in FP32, the adapter is roughly 6.9 MB; in BF16 it shrinks to about 3.5 MB, allowing many task-specific adapters to be hot-swapped against a single shared backbone.

## C. Additional Experiments

In this section we report supplementary studies referenced in the main text: (i) generalization to a non-biomedical regression task (STS-B); (ii) cross-architecture comparison with Llama-3.2-3B; (iii) ablation of the GLS-based fusion against naive inverse-variance weighting (IVW); and (iv) head-only and text-only ablations on FreeSolv.

### C.1. Generalization Beyond Biomedicine: STS-B

To verify that SPR-RAFT does not rely on biomedical domain priors, we evaluate it on STS-B (GLUE), a general-domain semantic textual similarity regression task with targets in $[0, 5]$. Following the same training recipe (Qwen3-4B, $m = 32$, 3 seeds, mean $\pm$ std), Table 6 shows that SPR-RAFT continues to outperform the RAFT baseline on a non-biomedical regression task.

*Table 6.* STS-B (GLUE) regression results. Mean ± standard deviation over 3 random seeds.

| Method | Pearson $r$ ↑ | Spearman $\rho$ ↑ |
|---|---|---|
| Zero-shot LLM | 0.61 ±0.03 | 0.62 ±0.03 |
| LoRA Fine-tune | 0.79 ±0.02 | 0.79 ±0.02 |
| CE Fine-tune | 0.82 ±0.02 | 0.82 ±0.02 |
| RAFT | 0.86 ±0.01 | 0.86 ±0.01 |
| **SPR-RAFT (ours)** | **0.89** ±0.01 | **0.89** ±0.01 |

*Table 7.* Cross-backbone comparison on TrialBench Duration (Phase I), MAE ↓. Mean over 3 random seeds.

| Backbone | RAFT | SPR-RAFT (ours) | Relative gain |
|---|---|---|---|
| Llama-3.2-3B | 1.92 ±0.10 | **0.93** ±0.05 | −51.6% |
| Qwen3-4B (default) | 0.94 ±0.06 | **0.70** ±0.03 | −25.5% |
| Gemma3-4B | 1.05 ±0.07 | **0.80** ±0.04 | −23.8% |
| Qwen3-8B | 0.88 ±0.05 | **0.68** ±0.03 | −22.7% |

## C.2. Cross-Architecture Backbones

Table 7 summarizes the comparison between RAFT and SPR-RAFT across four frozen backbones on TrialBench Duration (Phase I), with all other hyperparameters fixed. SPR-RAFT consistently improves over the corresponding RAFT baseline irrespective of model family or scale.

## C.3. GLS Fusion vs. Naive IVW and Single-Branch Ablations

The uncertainty-aware fusion in Section 3.4 adopts a Generalized Least Squares (GLS) form that accounts for the correlation $\rho$ between the two estimators. Table 8 compares this against a naive Inverse-Variance Weighting (IVW) fusion that assumes $\rho = 0$, as well as the two single-branch baselines (text-only and head-only) on FreeSolv. Naive IVW reaches a slightly lower RMSE than the head branch alone, but its negative log-likelihood (NLL) is substantially larger, indicating overconfident variance estimates that collapse under correlated errors. The GLS fusion both reduces RMSE and yields the best NLL, confirming that explicitly modeling the inter-branch correlation is necessary for calibrated predictions.

## C.4. Comparison with Modern Specialist Models on FreeSolv

To contextualize SPR-RAFT against modern molecular specialists, Table 9 compares against representative graph-based and chemistry-pretrained baselines. Despite using a generic LLM backbone with $\sim 0.04\%$ trainable parameters, SPR-RAFT achieves the lowest RMSE on FreeSolv. We emphasize that, in narrow homogeneous regimes such as pure SMILES regression, dedicated specialist models (e.g., chemistry-pretrained graph encoders) remain strong and often more efficient. The value of SPR-RAFT lies in providing a unified, parameter-efficient interface that performs competitively in such narrow regimes while extending naturally to heterogeneous biomedical evidence (clinical text, tabular labs, ICD codes) where specialists do not directly apply.

# D. Additional Figures

This section provides complementary visual diagnostics that are omitted from the main paper due to space. Figure 3 examines whether the uncertainty predicted by the [REG]-anchored head (via $s_\phi$) is informative about failures of the distribution-based text branch. Concretely, we correlate the head uncertainty with the text-branch regression error and additionally report a binned trend. The goal is not to claim perfect uncertainty calibration, but to validate that the fusion mechanism is most beneficial in regions where at least one branch exhibits higher uncertainty.

# E. Inference Cost and Scalability

This section discusses the inference-time overhead of SPR-RAFT and practical strategies for scaling to longer contexts and higher throughput settings.

*Table 8.* Ablation of fusion strategies on FreeSolv (RMSE / NLL, lower is better). Mean over 3 seeds.

| Variant | RMSE ↓ | NLL ↓ |
|---|---|---|
| Text branch only ($\mu_{\text{text}}$) | 0.531 ±0.03 | — |
| Head branch only ($\mu_\phi$) | 0.495 ±0.04 | 0.65 ±0.05 |
| Fusion w/ Naive IVW ($\rho = 0$) | 0.470 ±0.03 | 1.25 ±0.10 |
| **Fusion w/ GLS (ours)** | **0.455** ±0.04 | **0.43** ±0.04 |

*Table 9.* FreeSolv RMSE comparison with modern specialist baselines. Numbers for baselines are from the corresponding original publications.

| Method | FreeSolv RMSE ↓ |
|---|---|
| Graphormer (Ying et al., 2021) | 2.210 |
| Uni-Mol (Zhou et al., 2023) | 1.480 |
| Mol-LLM (Lee et al., 2025) | 0.874 |
| **SPR-RAFT (ours)** | **0.455** ±0.04 |

### E.1. Cost decomposition

SPR-RAFT produces two predictive distributions and then fuses them. At inference time, the total cost can be decomposed into:

1. **Backbone encoding** of the serialized input $x$ (shared by both branches).

2. **Distribution-based branch** (text branch): score a candidate set $C(x) = \{c_k\}_{k=1}^K$ of numeric strings using teacher-forced log-likelihood.

3. **Representation-based branch** (head branch): read out the [REG] anchor hidden state and apply a lightweight regression head.

4. **Fusion** of the two branches (negligible compute, dominated by a few scalar operations).

Let $L$ denote the total token length of the prompt (including the [REG] anchor and any task template), and let $S_k$ denote the token length of candidate string $c_k$ (typically small, e.g., a few digits plus punctuation). Let $\mathscr{F}(L)$ denote the cost of a single forward pass of the frozen backbone for a length-$L$ sequence under your implementation.

**Head-only baseline.** A head-only regressor that reads the [REG] hidden state requires a single backbone forward pass:

$$\text{Cost}_{\text{head}} \approx \mathscr{F}(L) + \mathcal{O}(dh), \tag{15}$$

where the second term is the regression head cost (matrix multiply on the anchor state), negligible relative to the backbone for typical $d$ and head hidden width $h$.

**Distribution branch.** Naively scoring $K$ candidates by appending each candidate string and running a full forward pass yields:

$$\text{Cost}_{\text{text, naive}} \approx \sum_{k=1}^K \mathscr{F}(L + S_k). \tag{16}$$

However, teacher-forced scoring admits substantial reuse. Since all candidates share the same prefix $x$, we can cache key-value (KV) states for the prefix and only process the short suffix $c_k$:

$$\text{Cost}_{\text{text, cached}} \approx \mathscr{F}(L) + \sum_{k=1}^K \mathscr{F}_{\text{suffix}}(S_k \mid L), \tag{17}$$

where $\mathscr{F}_{\text{suffix}}$ denotes the incremental decoding cost conditioned on a cached prefix of length $L$. In practice, $\mathscr{F}_{\text{suffix}}(S_k \mid L)$ is much smaller than $\mathscr{F}(L + S_k)$ because only $S_k$ new tokens are processed and attention attends to the cached prefix.

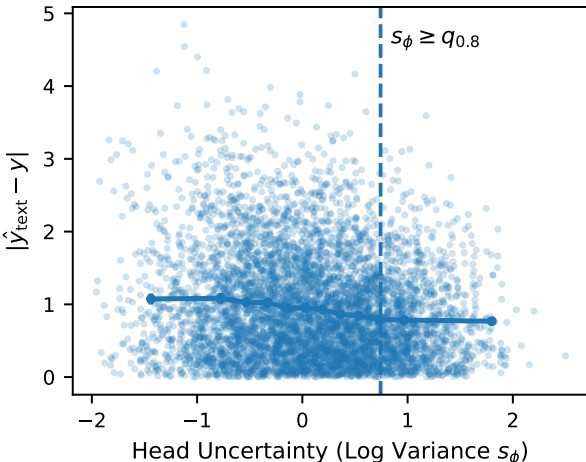

*Figure 3.* Effectiveness of uncertainty-aware fusion (diagnostic). The dashed line marks the high-uncertainty region of the regression head ($s_\phi \geq q_{0.8}$, top 20% by $s_\phi$). While the overall rank correlation between $s_\phi$ and the text-branch error is weak (Spearman $= -0.11$), the binned trend shows a clear tail behavior in this region where fusion is most critical. Uncertainty-aware fusion reduces MSE from 1.345 (text-only) to 1.129 (fusion), a 16.1% relative improvement.

**Total cost of SPR-RAFT inference.** With KV caching and batched candidate scoring, the total cost is:

$$\text{Cost}_{\text{SPR-RAFT}} \approx \mathscr{F}(L) + \sum_{k=1}^{K} \mathscr{F}_{\text{suffix}}(S_k \mid L) + \mathscr{O}(dh). \tag{18}$$

This highlights the key scaling knob: the candidate set size $K$, which increases compute approximately linearly in the total candidate suffix length $\sum_k S_k$, modulo batching efficiency.

### E.2. Scaling behavior with candidate set size and context length

**Candidate set size $K$.** If candidate suffix lengths are approximately constant ($S_k \approx \bar{S}$), then:

$$\text{Cost}_{\text{SPR-RAFT}} \approx \mathscr{F}(L) + K \cdot \mathscr{F}_{\text{suffix}}(\bar{S} \mid L). \tag{19}$$

Hence, increasing $K$ primarily increases incremental decoding work and memory bandwidth. On modern accelerators, the marginal cost per candidate often decreases when scoring candidates in a batch of size $K$, until memory saturation.

**Context length $L$.** Longer $L$ increases both the shared prefix cost $\mathscr{F}(L)$ and the per-candidate suffix cost, because incremental decoding attends to a longer cached prefix. Therefore, when $L$ is large (for example, long clinical notes or long tabular serialization), the relative overhead of the text branch can grow even if $S_k$ stays small.

**Practical implication.** For short to moderate $L$, SPR-RAFT overhead is typically dominated by $K$ and can be controlled by choosing $K$ or by using conditional compute (Section E.4). For very long $L$, reducing $K$ and using coarse-to-fine candidate refinement can be more important than micro-optimizing head computations.

### E.3. Empirical latency and throughput

We report end-to-end inference latency and throughput for different candidate set sizes $K$.

### E.4. Optimization strategies for scalable deployment

Below we summarize practical approaches to reduce overhead while preserving accuracy.

**1) Batched candidate scoring.** Score all candidates in a single batch of size $K$ with a shared cached prefix. This typically improves accelerator utilization and reduces kernel launch overhead relative to sequential scoring.

*Table 10.* Inference cost of SPR-RAFT as a function of candidate set size $K$. Latency is per example. Throughput is measured with a fixed batch size and includes preprocessing and fusion.

| Setting | $K$ | Latency (ms) $\downarrow$ | Throughput (ex/s) $\uparrow$ | Peak VRAM (GB) $\downarrow$ |
|---|---|---|---|---|
| Head-only (no text branch) | 0 | 18.0 | 55.0 | 9.2 |
| SPR-RAFT (cached, batched) | 8 | 22.5 | 44.0 | 10.1 |
| SPR-RAFT (cached, batched) | 16 | 27.8 | 36.0 | 11.3 |
| SPR-RAFT (cached, batched) | 32 | 39.5 | 25.0 | 13.8 |

*Table 11.* Scaling with context length $L$. Report the same $K$ and batch size across rows.

| Context length $L$ | Latency (ms) $\downarrow$ | Throughput (ex/s) $\uparrow$ | Notes |
|---|---|---|---|
| 512 | 27.8 | 36.0 | KV caching enabled |
| 1024 | 36.2 | 27.5 | same $K$ and batch |
| 2048 | 58.0 | 17.2 | attention dominates |

**2) Prefix KV caching and suffix-only decoding.** Cache KV states for the shared prefix once, then perform incremental decoding for each candidate suffix. This converts $K$ full forward passes into 1 full pass plus $K$ short suffix passes.

**3) Coarse-to-fine candidate sets.** Use a small coarse grid $C_0(x)$ (for example, $K = 8$) to obtain a rough estimate, then refine around the most likely region with a local grid $C_1(x)$ (for example, another $K = 8$). This keeps the worst-case cost manageable while maintaining resolution where it matters.

**4) Conditional compute via uncertainty gating.** Use the predicted uncertainty from either branch to decide whether to invoke the distribution branch. For example, skip candidate scoring when the head branch is confident:

$$\text{invoke text branch} \iff \sigma_\phi(x) > \tau, \tag{20}$$

where $\tau$ is a threshold tuned on the validation set. This yields an adaptive compute-accuracy tradeoff, especially effective when many inputs are easy.

**5) Token-level and vocabulary-level pruning.** Since candidate strings are numeric, candidate scoring can be restricted to the log-probabilities of the candidate tokens rather than full-vocabulary outputs, when supported by the inference stack. This can reduce memory bandwidth and improve throughput.

**6) Quantization and compilation.** Because the backbone is frozen, post-training quantization (for example, int8 or int4 where acceptable) and compiler optimizations (kernel fusion, attention optimizations) can reduce latency without changing SPR-RAFT training.

**7) Parallelism across candidates.** For large $K$ or high throughput, candidates can be partitioned across devices or streams and reduced by log-sum-exp aggregation. This is most useful when batch size is already large and memory becomes the bottleneck.

