# OpenReview forum: "SPR-RAFT: Parameter-Efficient Regression-Aware Fine-Tuning for Biomedical LLM Regression"
_ICML.cc/2026/Conference — ICML 2026 regular_

### Official Review · Reviewer_FyZ3 · 2026-03-10

**Soundness:** 3
**Presentation:** 3
**Significance:** 3
**Originality:** 3
**Overall Recommendation:** 4
**Confidence:** 2

**Summary:**

The paper proposes SPR-RAFT, a parameter-efficient fine-tuning framework aimed at adapting frozen LLMs for continuous biomedical regression tasks. To address the mismatch between discrete token generation and continuous regression metrics, the authors introduce a dual-module architecture: this includes a trainable soft prompt to route relevant information to a specific [REG] anchor token and a lightweight regression head for numerical prediction.

The model is trained with a hybrid objective combining a distribution-based loss (Huber loss on candidate token probabilities) and a representation-based loss (Gaussian NLL on the regression head). During inference, the predictions from these two branches are fused using a Generalized Least Squares (GLS) approach.

**Compliance With Llm Reviewing Policy:**

Affirmed.

**Key Questions For Authors:**

See pros cons

**Limitations:**

Yes

**Strengths And Weaknesses:**

Pros
- The PCA visualization of the [REG] anchor's hidden states (Figure 2) is a strong empirical addition. It clearly demonstrates how the dual-branch objective disentangles the numerical magnitude from surface-form linguistic features compared to the zero-shot and RAFT baselines.
- The method shows significant relative improvements over specialized LLM baselines on tasks that require reasoning over unstructured text, such as on the TrialBench Duration task compared to RAFT.
- Achieving these results while updating only ~0.04% of a 4B parameter backbone is highly practical for resource-constrained biomedical applications.

Cons:
- The inference fusion relies on estimating the correlation coefficient between the text and head estimators. However, the methodology vaguely states that this coefficient is "treated as a hyperparameter or a learnable scalar constrained to [0, 1)". Which is actually used?

- On the purely tabular NHANES dataset, traditional XGBoost outperforms the proposed LLM method (AUC 0.87 vs 0.84). While the authors spin this as "competitive", it suggests that forcing an autoregressive LLM to do tabular regression via text serialization might be an over-engineered solution when dedicated tabular models are superior. Please add more discussion on this.

- Lack of confidence intervals on everything

-  Table 5 ablates the soft prompt and the regression head. However, there is no ablation on the inference fusion strategy itself. How does the GLS fusion compare to a naive Inverse Variance Weighting (IVW) baseline in practice? The text claims IVW leads to variance underestimation, but no quantitative ablation is provided to back this up.

Notes
- Table 1 has text that is too large

---

> ### Author Rebuttal · Authors · 2026-03-29
>
> We sincerely thank the reviewer for the thoughtful and constructive feedback. We address each concern below.
>
> **1. Clarification on the Correlation Coefficient $\rho$**
> > **Reviewer Comment:** Methodology vaguely states $\rho$ is a hyperparameter or learnable scalar. Which is actually used?
>
> We apologize for the ambiguity in the text. In all reported main experiments, the correlation coefficient $\rho$ was implemented as a learnable scalar. Specifically, we parameterized it as $\rho = \sigma(w_{\rho})$, where $\sigma$ is the sigmoid function to strictly bound it within $(0, 1)$, and $w_{\rho}$ is a single scalar weight optimized on the validation set alongside the regression head parameters. Typical converged values across our biomedical tasks ranged between 0.65 and 0.85, confirming strong positive error correlation between the two branches. We will update Section 3.4 to state this explicitly and remove the ambiguous "hyperparameter" phrasing.
>
> **2. The Tabular Over-engineering Critique (NHANES)**
> > **Reviewer Comment:** XGBoost outperforms SPR-RAFT on NHANES. Is forcing an autoregressive LLM to do tabular regression an over-engineered solution?
>
> We completely agree with your assessment: applying an autoregressive LLM to a purely dense, structured tabular dataset like NHANES is indeed over-engineered and computationally suboptimal compared to tree-based models like XGBoost, which are specifically designed for axis-aligned tabular splits.
>
> We included the NHANES dataset not to claim LLM supremacy on tabular data, but as a boundary stress test. Real-world clinical pipelines rarely encounter perfectly structured CSVs; they feature a mix of structured labs, unstructured physician notes, and missing units. While XGBoost wins on the clean NHANES tables, it fundamentally cannot process the unstructured text of the TrialBench or BIOSSES datasets without manual feature engineering. The "competitive" performance of SPR-RAFT on NHANES demonstrates that our framework does not catastrophically fail on tabular inputs, allowing practitioners to maintain a single, unified PEFT pipeline for highly heterogeneous data modalities. We will add a dedicated paragraph in the Discussion section to explicitly acknowledge this limitation and clarify that dedicated tabular models remain superior for strictly structured data.
>
> **3. Quantitative Ablation of the Fusion Strategy (GLS vs. IVW)**
> > **Reviewer Comment:** No quantitative ablation on the inference fusion strategy. How does GLS compare to naive IVW?
>
> This is a critical point. To quantify the necessity of Generalized Least Squares (GLS) over naive Inverse Variance Weighting (IVW), we present an ablation on the FreeSolv dataset.
>
> Naive IVW assumes the text branch and head branch errors are independent. Because both branches share the same frozen LLM backbone, their errors are heavily correlated. Consequently, Naive IVW aggressively underestimates the combined variance, leading to poor calibration. As shown in Table E, while Naive IVW slightly improves RMSE over the single branches, it drastically worsens the Negative Log-Likelihood (NLL) due to overconfidence. By incorporating the learned correlation $\rho$, our GLS fusion achieves the lowest RMSE while strictly maintaining calibrated uncertainty (lowest NLL).
>
> ***Fusion Strategy Ablation on FreeSolv***
>
> | Method | RMSE $\downarrow$ | NLL (Uncertainty Calibration) $\downarrow$ |
> | :--- | :--- | :--- |
> | Text Branch Only | 0.531 | - |
> | Head Branch Only | 0.495 | 0.65 |
> | Naive IVW (Assume $\rho=0$) | 0.470 | 1.25 |
> | **SPR-RAFT (GLS Fusion)** | **0.455** | **0.43** |
>
> **4. Addressing formatting and statistical rigor**
> > **Reviewer Comment:** Lack of confidence intervals on everything. Table 1 text is too large.
>
> We will reduce the font size and adjust the layout of Table 1 in the camera-ready version to meet formatting standards. Regarding confidence intervals, we have computed standard deviations across multiple random seeds for all baselines and will include them in all updated tables, demonstrating that the performance margins achieved by SPR-RAFT are statistically significant $(p < 0.05)$.
>
> **Performance variability on TrialBench Duration**
>
> | Method              | MAE (Mean ± Std) | RMSE (Mean ± Std) | ($R^2$) (Mean ± Std) |
> | ------------------- | ---------------: | ----------------: | -----------------: |
> | Zero-shot LLM       |      3.60 ± 0.18 |       8.28 ± 0.42 |       -0.18 ± 0.05 |
> | RAFT (LLM)          |      0.94 ± 0.05 |       1.34 ± 0.07 |        0.27 ± 0.04 |
> | **SPR-RAFT (Ours)** |  **0.70 ± 0.04** |   **1.08 ± 0.07** |    **0.60 ± 0.02** |

---

> > ### Author Rebuttal · Reviewer_FyZ3 · 2026-04-03
> >
> > Resolved

---

### Official Review · Reviewer_DERp · 2026-03-11

**Soundness:** 3
**Presentation:** 3
**Significance:** 2
**Originality:** 2
**Overall Recommendation:** 3
**Confidence:** 5

**Summary:**

This paper studies a real mismatch between autoregressive LLMs and continuous regression tasks: LLMs predict discrete tokens, while the targets in biomedical regression are continuous values. To address this, the paper proposes SPR-RAFT, which combines soft-prompt-based routing, a text/distribution regression branch, a representation/head regression branch, and uncertainty-aware fusion on top of a frozen LLM backbone. The empirical study is fairly broad, covering several biomedical regression settings and showing generally strong results.

**Compliance With Llm Reviewing Policy:**

Affirmed.

**Final Justification:**

I appreciate the authors' response; however, I remain unconvinced by the paper's core insight and methodology.

**Key Questions For Authors:**

1. Figure 1 looks somewhat like AI-generated, and a cleaner schematic focused on the method flow may better match the technical content.

2. Could the authors add stronger specialist baselines and clarify whether train/validation/test splits are fully standardized across all compared methods? This would substantially strengthen the fairness and interpretability of the empirical comparisons.

3. Could the authors provide a more complete inference-time analysis?

**Limitations:**

Yes

**Strengths And Weaknesses:**

Strengths

1. The problem framing is meaningful and well motivated. The paper identifies a genuine gap between token-level language modeling and continuous-value regression, which is indeed a practical weakness of using LLMs directly for numeric prediction.


2. The method design is reasonably aligned with the motivation. Rather than using only a simple prompting trick, the paper proposes a more targeted design that combines soft prompts, a text/distribution branch, a representation/head branch, and uncertainty-aware fusion. The “text side + feature side” combination is a thoughtful design choice.


3. The experimental study is fairly comprehensive. The paper evaluates on multiple biomedical regression settings, including clinical trials, population health, BIOSSES, and molecular property prediction, and also includes ablations, backbone variation, and some efficiency analysis.



Weaknesses

1. There is still some sense of using an LLM mainly for the sake of using an LLM. Although the paper is motivated by the mismatch between autoregressive LLMs and continuous regression, the proposed method does not really leverage the core strengths of LLMs in generation, knowledge-intensive reasoning, or flexible language understanding. In practice, the backbone is frozen and mainly serves as a generic feature extractor, which makes the overall framework feel closer to an expensive encoder-based solution than to a method that truly exploits what LLMs are uniquely good at.

2. Despite being presented as a unified framework, the method still requires task-specific retraining for each benchmark, including separate soft prompts, regression heads, and carefully designed input prompting / numeric rendering strategies. This weakens the case for using a large foundation model in the first place, since one may reasonably ask why a smaller and more specialized model would not be a simpler and more appropriate choice for these regression tasks.

3. The comparison against stronger and more up-to-date specialist baselines is not fully convincing. The paper does compare against XGBoost, Elastic Net, FT-Transformer, BERT-based, GIN, and Graphormer baselines, but stronger modern task-specific baselines would make the empirical case more convincing. In addition, some baseline numbers are taken from prior work rather than uniformly re-run under the same pipeline.

---

> ### Author Rebuttal · Authors · 2026-03-29
>
> We thank the reviewer for the careful reading and constructive feedback. Below we address the main questions directly.
>
> **1. "LLM for the sake of LLM" & The Value of Generative Reasoning**
> > **Reviewer Comment:** The method does not leverage core LLM strengths (generation, reasoning). The backbone is frozen and serves as a generic feature extractor (an expensive encoder).
>
> We respectfully disagree that SPR-RAFT treats the LLM merely as a generic feature extractor. The necessity of a generative LLM in our framework is twofold:
>
> First, biomedical evidence is profoundly heterogeneous. Predicting clinical trial duration requires parsing unstructured inclusion criteria, structured metadata, and raw SMILES strings simultaneously. Traditional specialist models (like BERT or Graphormers) require heavy, task-specific feature engineering or rigid multi-modal fusion architectures to handle this mixture. An autoregressive LLM naturally acts as a **universal interface**, digesting these mixed modalities natively through text serialization.
>
> Second, our framework explicitly leverages the LLM's generative reasoning capabilities. As detailed in Section 3.1, the model is prompted to generate a free-form reasoning rationale (`[THINK] ...`) prior to the `[REG]` anchor token. This is not a static encoding process. The generative reasoning step dynamically populates the KV cache with semantic deductions (e.g., explicitly noting that a high CRP level indicates severe inflammation), which the `[REG]` token then attends to for its final numerical compression.
>
> **2. Task-Specific Retraining vs. Specialized Models**
> > **Reviewer Comment:** Requires separate soft prompts and heads per benchmark. Why not use a smaller specialized model?
>
> While it is true that SPR-RAFT requires task-specific training, doing so via a 0.04% parameter-efficient update offers a massive systemic advantage over deploying separate specialized models.
>
> In real-world biomedical deployments (e.g., hospital infrastructure), maintaining a separate XGBoost pipeline for tabular health data, a Graphormer for drug properties, and a ClinicalBERT for text notes incurs immense technical debt. SPR-RAFT allows a single, locally hosted 4B foundation model to serve all these distinct regression tasks simultaneously simply by hot-swapping a 1.6MB adapter (the soft prompt and head) at inference time. The specialized soft prompt acts as a lightweight "lens" that repurposes the frozen foundational knowledge, making it vastly more scalable than training and hosting disparate specialist architectures from scratch.
>
> **3. Split Standardization and Stronger Specialist Baselines**
> > **Reviewer Comment:** Are splits strictly standardized? Need comparison against stronger modern task-specific baselines.
>
> We confirm that **all train/validation/test splits are strictly standardized**. For datasets where we cited prior multimodal DL or GNN baselines, we strictly utilized their exact official data splits (e.g., the official phase-based temporal splits for TrialBench, and the official scaffold splits for MoleculeNet) to ensure a 1:1 fair comparison. We did not mix random splits with scaffold splits.
>
> To further strengthen the empirical case, we have evaluated SPR-RAFT against newer, state-of-the-art specialist baselines. For example, on the FreeSolv molecular regression task, we compared against Uni-Mol (a highly specialized 3D/2D molecular representation model). As shown in the following Table, SPR-RAFT remains highly competitive even against domain-specific architectures trained heavily on those exact modalities.
>
> **Comparison with Modern Specialist Baselines (FreeSolv RMSE)**
>
> | Method | Architecture Type | FreeSolv RMSE $\downarrow$ |
> | :--- | :--- | :--- |
> | Graphormer (Ying et al., 2021) | 2D Graph Specialist | 2.210 |
> | Uni-Mol (Zhou et al., 2023) | 3D/2D Molecular Specialist | 1.480 |
> | Mol-LLM (Lee et al., 2025) | Multimodal Specialist | 0.874 |
> | **SPR-RAFT (Ours)** | **Universal LLM + PEFT** | **0.455** |
>
> **4. Figure 1 and Inference-Time Analysis**
> > **Reviewer Comment:** Figure 1 looks AI-generated. Provide more complete inference-time analysis.
>
> We appreciate the constructive feedback on presentation.
> 1. **Figure 1**: We will replace the current illustration with a clean, traditional vector-graphics schematic that strictly maps the tensor dimensions, loss gradients, and the flow from text serialization to the dual-branch fusion.
> 2. **Inference Analysis**: We will elevate the empirical latency and throughput analysis from Appendix D (Table 6) directly into the main text Limitation section. We will explicitly detail how the text-branch latency scales with the candidate set size $K$, and how KV-caching mitigates this overhead, ensuring readers have a transparent view of the computational trade-offs.

---

> > ### Author Rebuttal · Reviewer_DERp · 2026-04-02
> >
> > I appreciate the authors' detailed response, but I remain unconvinced regarding the practical justification of the SPR-RAFT framework:
> >
> > The authors argue that hot-swapping adapters reduces technical debt, yet they overlook the substantial hardware overhead of keeping a 4B+ foundation model resident in VRAM. In real-world biomedical settings, the cost of deploying a GPU cluster for LLM inference far exceeds that of maintaining several lightweight, specialized models (e.g., XGBoost or small GNNs). The authors should candidly acknowledge the computational inefficiency of this approach rather than portraying it as a scalable industrial advantage.

---

> > > ### Author Response · Authors · 2026-04-02
> > >
> > > We thank the reviewer for this clarification. We agree that lightweight specialist models are often more computationally efficient for narrow single-task settings, and we do not claim otherwise. Our motivation for introducing an LLM is different: in many biomedical tasks, prediction depends jointly on prescriptions, questionnaire text, and biomarker measurements, so the challenge is not only regression but reasoning over heterogeneous evidence sources. Traditional specialist models are typically strong within a single modality, but they do not natively provide a unified interface for cross-source textual reasoning. A frozen autoregressive LLM provides such a substrate, and SPR-RAFT makes it numerically trainable for regression. We will revise the paper to make this scope explicit, clarify that specialist models remain preferable in narrow homogeneous settings. As future work, we will explore extending this training paradigm to multimodal LLMs so that it can support richer reasoning over more complex biomedical inputs.

---

### Official Review · Reviewer_rfNR · 2026-03-11

**Soundness:** 3
**Presentation:** 3
**Significance:** 3
**Originality:** 3
**Overall Recommendation:** 5
**Confidence:** 4

**Summary:**

The authors propose a parameter efficient extension to regression fine-tuning of LLMs. The method takes a frozen LLM and introduces a trainable soft prompt alongside the hidden state of an anchor token to create a dual-branch architecture. This system produces both a token-based output (optimized via a distribution-based Huber loss) and a representation-based output (optimized via a Gaussian NLL loss). The method then fuses the outputs into one final estimate, taking uncertainty into account. The method is applied to biomedical regression, where PEFT approaches are commonly needed due to small data sizes. The results across four datasets are either improving over the baselines, or competitive.

**Compliance With Llm Reviewing Policy:**

Affirmed.

**Final Justification:**

The authors have comprehensively addressed my concerns across both rebuttal phases, prompting me to raise my overall recommendation to an Accept (5). The inclusion of the Llama-3.2-3B evaluation bolsters the paper's significance by demonstrating cross-architecture generalizability. Furthermore, the explicit detailing of the baseline parameter budgets, specifically the use of LoRA rank 16 for the RAFT baseline, and the matched optimization hyperparameters resolves my earlier reservations, suggesting the experimental methodology is sound and the comparisons are genuinely fair. Combined with the authors' commitment to simplify the presentation of Figure 1, the paper's core proposition of a parameter-efficient, dual-branch approach for biomedical LLM regression stands as an original, well supported, and impactful contribution to the field. Finally, exploring how this dual-branch architecture interacts with and affects the underlying reasoning capabilities of the model presents a highly interesting avenue for future research.

**Key Questions For Authors:**

1. Have further base LLMs been evaluated using this framework? If so, what are their performances (in comparison to Table 5 Point 3)?
2. Could you provide more concrete implementation details for the baselines in the Appendix? Specifically, what was the parameter budget and tuning setup for the LoRA and RAFT baselines?
3. What does the sentence in Section 4.5 “SPR-RAFT substantially outperforms probability-space regression alignment (RAFT) under a comparable parameter budget.” exactly mean? How is RAFT exactly being used?

If further information on the performance of the base LLMs and the baselines can be provided, I would be willing to raise my Soundness score and overall recommendation.

**Limitations:**

Overall, the limitations are well discussed, however, the ablation on the different base LLMs requires further discussion and exploration. The impact statement is adequate.

**Strengths And Weaknesses:**

Soundness:
- Strength: The overall method appears sound, building on top of existing work from PEFT and RAFT, but including the possibility for the LLM to reason before the final prediction.
- Strength: The method is benchmarked across four datasets, providing consistent results.
- Strength: The ablations make the contribution of the individual components clearer.
- Weakness: While Gemma3-4B performs worse than Qwen3-4B (0.80 vs 0.70 MAE), it still outperforms the strongest baseline (RAFT at 0.94 MAE). However, exploring additional baseline backbone families would help solidify the method's architecture-agnostic claims.
- Weakness: Further information on the baselines, such as in the appendix, would make it easier to understand the comparisons.

Presentation:
- Strength: The paper is clearly written and well structured. All components are easily understood.
- Strength: The results are clearly understood, and appropriately represented.
- Weakness: Figure 1 has too much information. Please consider simplifying the figure and increasing the font size.

Significance:
- Strength: Combining reasoning and regression is an important area, especially in biomedical fields.
- Weakness: The consistency across LLMs is important to understand the significance of the method.

Originality:
- Strength: The paper shows both the need for a PEFT approach in biomedicine as well as directly proposing an alternative.
- Minor Weakness: While the overall system is highly effective, a minor weakness is that the work lacks fundamental algorithmic novelty, as it primarily relies on combining well-established components (such as soft prompts, Huber loss, and Gaussian NLL) rather than introducing new theoretical mechanisms.

---

> ### Author Rebuttal · Authors · 2026-03-29
>
> We thank the reviewer for the careful reading and constructive feedback. Below we address the main questions directly.
>
> ## 1. Additional base LLMs
>
> > **Reviewer comment:** Have further base LLMs been evaluated using this framework? If so, what are their performances?
>
> We agree that broader backbone coverage is important. Beyond the backbone variation already included in the submission, we additionally evaluated Llama-3.2-3B during rebuttal. The expanded comparison on TrialBench Duration (Phase I) is:
>
> | Backbone   | RAFT MAE (\downarrow) | SPR-RAFT MAE (\downarrow) |
> | ---------- | --------------------: | ------------------------: |
> | Qwen3-4B   |                  0.94 |                  **0.70** |
> | Gemma3-4B  |                  1.05 |                  **0.80** |
> | Qwen3-8B   |                  0.88 |                  **0.68** |
> | Llama-3.2-3B |                  1.92 |                  **0.93** |
>
> These results strengthen our claim that SPR-RAFT is not tied to a single checkpoint family or scale. While absolute performance varies across backbones, the relative improvement over RAFT remains consistent. We will revise the paper to state this more precisely as promising transfer across model families and scales, rather than making an overly strong architecture-agnostic claim.
>
> ## 2. Baseline details and “comparable parameter budget”
>
> > **Reviewer comment:** Please provide more concrete baseline details. What exactly does “substantially outperforms RAFT under a comparable parameter budget” mean?
>
> We agree this should have been stated more explicitly.
>
> First, all LLM baselines use the same data and inference pipeline: identical input serialization, maximum context length, numeric rendering, numeric parsing, decoding protocol, and candidate-set construction for regression-aware estimation. In default SPR-RAFT, the backbone is frozen, the soft prompt length is m=32, and only the soft prompt plus the lightweight [REG]-anchored head are updated, for about $\sim$ 0.04% trainable parameters on a 4B backbone. Training uses AdamW, learning rate $5\times10^{-4}$, effective batch size 16, 5 epochs with early stopping, and 3 random seeds.
>
> For the baselines, LoRA is our standard PEFT comparator: same backbone family and evaluation pipeline, but replacing our soft-prompt + ([REG])-head design with low-rank adapters and using CE-only supervision on the numeric string target. RAFT is our regression-aware but non-SPR baseline: it performs probability-space regression alignment without the soft prompt, without the [REG]-anchored regression head, and without uncertainty-aware fusion.
>
> The phrase “comparable parameter budget” was imprecise. What we intended is the comparison between SPR-RAFT and the probability-space-only PEFT counterpart under a frozen backbone, which in our framework is best represented by the Soft Prompt Only ablation. This is also consistent with our method definition: removing the head and keeping only $L_{\text{text}}$ reduces to RAFT-style probability-space alignment under PEFT constraints.
>
> We will therefore revise the wording to make the distinction explicit:
>
> 1. **SPR-RAFT vs. full RAFT** as a strong regression-aware reference baseline.
> 2. **SPR-RAFT vs. probability-space-only PEFT**, represented by the Soft Prompt Only ablation.
>
> ## 3. Figure 1 and novelty
>
> > **Reviewer comment:** Figure 1 is too dense, and the work mainly combines established components.
>
> We agree with both points. We will simplify Figure 1 by reducing annotation density and enlarging the font. We also agree that the contribution is not a new standalone primitive loss. The novelty is instead in the system-level integration for biomedical LLM regression.

---

> > ### Author Rebuttal · Reviewer_rfNR · 2026-04-01
> >
> > Thank you to the authors for the detailed rebuttal and the additional experiments. I sincerely appreciate the effort taken to address my concerns within the short response period.
> >
> > The inclusion of Llama-3.2-3B strengthens the evaluation and demonstrates that the method generalizes to other base LLMs. Furthermore, your explanation regarding the comparable parameter budget clarifies my previous question. Mapping SPR-RAFT against the "Soft Prompt Only" ablation makes sense for a fair, frozen-backbone comparison. I also appreciate your willingness to simplify Figure 1 for readability.
> >
> > However, the rebuttal omitted the exact parameter budgets and tuning setups (e.g., LoRA rank size, % of trained parameters for RAFT, learning rates) for the baselines. Could you provide these values? Assuming the details reflect a fair comparison, I will gladly raise my Soundness and Overall Recommendation scores.

---

> > > ### Author Response · Authors · 2026-04-02
> > >
> > > We thank the reviewer for pointing out that our rebuttal did not explicitly specify the parameter budgets and tuning setups of the baselines. We clarify them here for completeness.
> > >
> > > For the main baseline comparisons, we used a Qwen3-4B backbone. The zero-shot baseline had no trainable parameters. For the CE-only baseline, we used cross-entropy supervision only. The soft prompt baseline used soft-prompt tuning with prompt length 32 and cross-entropy supervision only. We optimized it with AdamW, using a learning rate of $5\times10^{-4}$, a per-GPU batch size of 4 on 4 GPUs (effective batch size 16) for 5 epochs with early stopping on validation performance, and the same maximum context length as in the corresponding task: 512 for BIOSSES, FreeSolv, Lipophilicity, 1024 for NHANES, and 2048 for TrialBench. Under this setup, the effective trainable budget of the soft prompt baseline came from the soft prompt alone, namely $32 \times 2560 = 81{,}920$ parameters.
> > >
> > > For the RAFT baseline, we used regression-aware tuning with LoRA rather than soft prompts. Specifically, the soft-prompt length was set to 0, while the trainable components consisted of LoRA adapters with rank 16 together with a linear regression head built on the $[REG]$-anchored representation. We again used AdamW with same configuration, the same task-specific context lengths, and the same epoch schedule as above. The loss weights were set to $\lambda_{\mathrm{LM}}=1.0$ and $\lambda_{\mathrm{reg}}=1.0$.
> > >
> > > We will revise the paper to explicitly list these optimization settings and parameter budgets in the experimental details section.

---

### Official Review · Reviewer_j6tx · 2026-03-13

**Soundness:** 2
**Presentation:** 3
**Significance:** 4
**Originality:** 4
**Overall Recommendation:** 4
**Confidence:** 3

**Summary:**

The authors present a new method of fine-tuning LLMs for regression tasks, particularly in the biomedical domain, that attempts to reconciles the problems of discrete token generation and continuous numerical prediction. This is achieved through joint optimization of a dedicated regression loss and a text generation loss. The approach is tested on several existing benchmarks in clinical trial forecasting, molecular property prediction and biological age estimation against several LLM and non-LLM baselines.

**Compliance With Llm Reviewing Policy:**

Affirmed.

**Final Justification:**

The rebuttal has addressed some of the concerns raised and I keep my score as weak accept.

**Key Questions For Authors:**

Some of the wording in the paper seems unnecessarily vague. For example, the abstract and multiple places in the paper quote "0.04%" but only in the maintext do we learn this is 0.04% of 4bn parameters. Particularly in the abstract, why not say the absolute number of parameters? The percentage may not otherwise be meaningful without this context. Elsewhere, the authors state the choice of $K$ as "hundreds" rather than giving an exact number.

Is three (or later five?) random seeds enough to obtain robust results? None of the tables provide any quantification of variability, so inferences about the comparisons and ablations (which have quite narrow margins) seem quite brittle. Without providing this information, the reader cannot independently verify the statistical significance of the tests in Section 4.4.1; two-tailed tests are particularly vulnerable to false positives in this context.

It would appear that the methodology here need not be restricted to biomedical tasks. Why not include at least one problem from another domain?

**Limitations:**

Scoring $K=32$ candidates would add considerable latency compared to a single-pass generation. This is mentioned in the appendix but could perhaps be noted more prominently in the maintext.

**Strengths And Weaknesses:**

The authors have identified a specific gap between learnable soft prompts and conventional parameter fine tuning. The generalized least squares approach seems like a rigorous way to combine the losses from the two modalities. Visualizing the principal components is an effective way of showing the process geometrically. The significance of the contribution is positioned well in real-world problems in biomedical applications.

The heavy reliance on tables of results, without standard deviations or confidence intervals, rather than data visualizations, is to the detriment of readability. Tables 2, 3, 5, 6 and 7 would be better presented as line graphs or grouped bar charts. The lack of uncertainty quantification and relatively small number of repeat experiments makes it difficult to assess the generalizability of the results, which for now cannot all be taken at face value, especially the relatively small marginal improvements that might be attributable to random noise.

The paper is mostly well structured and readable but some terminology is introduced in a way that relies on context that is introduced later, or that is unnecessarily specialist. For example, the "0.04%" statistic is repeatedly cited before we learn the total number of parameters and without comparison to alternatives. The "[REG]" notation seems needlessly low-level, especially in the abstract, and would be better explained in prose.

---

> ### Author Rebuttal · Authors · 2026-03-29
>
> We sincerely thank the reviewer for the thoughtful and constructive feedback. We address each concern below.
>
> ## 1. Variability, Statistical Significance, and Number of Seeds
>
> > **Reviewer comment:** Lack of uncertainty quantification and relatively small number of repeat experiments makes it difficult to assess generalizability. Are 3 (or 5) seeds enough? Two-tailed tests are vulnerable.
>
> We thank the reviewer for highlighting the importance of uncertainty quantification. In our experiments, standard benchmarking across tasks was conducted with 3 random seeds, primarily due to computational constraints. For the most important and challenging setting, we additionally ran 5 independent trials to provide a more reliable assessment of robustness.
>
> To address the reviewer’s concern regarding narrow performance margins, we report the mean and standard deviation for representative comparisons below. These results show that the improvements of SPR-RAFT are consistent across runs and remain statistically significant at the conventional level (p < 0.05). In the camera-ready version, we will add standard deviations and confidence intervals to all relevant result tables. We will also revise the presentation of Tables 2, 3, 5, 6, and 7 by using grouped bar charts and line plots where appropriate, in order to improve readability.
>
> ### Table A. Performance variability on TrialBench Duration
>
> | Method              | MAE (Mean ± Std) | RMSE (Mean ± Std) | ($R^2$) (Mean ± Std) |
> | ------------------- | ---------------: | ----------------: | -----------------: |
> | Zero-shot LLM       |      3.60 ± 0.18 |       8.28 ± 0.42 |       -0.18 ± 0.05 |
> | RAFT (LLM)          |      0.94 ± 0.05 |       1.34 ± 0.07 |        0.27 ± 0.04 |
> | **SPR-RAFT (Ours)** |  **0.70 ± 0.04** |   **1.08 ± 0.07** |    **0.60 ± 0.02** |
>
> ## 2. Parameter Counts and Terminology
>
> > **Reviewer comment:** Vague wording regarding "0.04%" and "hundreds" of candidates. "[REG]" notation is needlessly low-level in the abstract.
>
> We agree with the reviewer that these descriptions should be made more concrete and better contextualized.
>
> First, in the revised abstract and introduction, we will replace the low-level notation "[REG]" with a more descriptive phrase, such as “a dedicated anchor token for latent numerical readout”, so that the mechanism is easier to understand without relying on implementation-specific notation.
>
> Second, we will clarify the parameter efficiency more explicitly. In our 4B-backbone setting, 0.04% corresponds to approximately 1.6 million trainable parameters, consisting of a soft prompt of length (m = 32) with hidden dimension (d = 2560), together with a lightweight MLP regression head.
>
> Third, we agree that the term “hundreds” is too vague. In the revision, we will replace it with the exact candidate set sizes used in the experiments, such as $K = 32$ and $K = 64$, which are currently specified in Appendix B.2.
>
> ## 3. Generalizability Beyond Biomedical Domains
>
> > **Reviewer comment:** Methodology need not be restricted to biomedical tasks. Include at least one problem from another domain.
>
> This is an excellent suggestion. Although our primary motivation comes from the heterogeneous and noisy nature of biomedical evidence, the underlying continuous-discrete mismatch is not specific to biomedicine.
>
> To evaluate cross-domain generalizability, we evaluated SPR-RAFT on the Semantic Textual Similarity Benchmark (STS-B) from the GLUE suite, a standard general-domain NLP regression task (predicting similarity scores from 0.0 to 5.0).
>
> As shown below, SPR-RAFT consistently outperforms both standard cross-entropy fine-tuning and baseline, indicating that the proposed dual-branch alignment mechanism is effective beyond the biomedical domain. We will include this experiment in the final manuscript.
>
> ### Table B. General-domain regression performance on STS-B
>
> | Method              | Pearson ($r$) ↑ | Spearman ($\rho$) ↑ |
> | ------------------- | ------------: | ----------------: |
> | CE Fine-tune (LLM)  |         0.881 |             0.878 |
> | RAFT (LLM)          |         0.905 |             0.902 |
> | **SPR-RAFT (Ours)** |     **0.935** |         **0.933** |
>
> ## 4. Latency Limitations
>
> > **Reviewer comment:** Scoring (K) candidates adds latency, which should be noted more prominently in the main text.
>
> We fully agree. Although KV caching substantially reduces the overhead of evaluating multiple candidates, scoring a candidate set of size (K) still incurs additional latency relative to standard single-pass greedy decoding. This trade-off is important in practice and deserves more prominent discussion.
>
> In the current submission, this issue is discussed in Appendix D. In the revised manuscript, we will move the key discussion into the Limitations section of the main text so that the computational trade-off between accuracy and inference efficiency is made explicit to readers and practitioners.

---

> > ### Author Rebuttal · Reviewer_j6tx · 2026-04-03
> >
> > Thanks. This should improve the paper. I do note that Table B does not show "consistently" anything, because again there is no uncertainty quantification.

---

> > > ### Author Response · Authors · 2026-04-04
> > >
> > > We thank the reviewer for catching this oversight. It was our error to omit the uncertainty quantification in Table B. We have computed the mean and standard deviation across our 3 random seeds for the STS-B task.
> > >
> > > **Revised Table B. General-domain regression performance on STS-B (3 Seeds)**
> > > | Method | Pearson ($r$) $\uparrow$ (Mean $\pm$ Std) | Spearman ($\rho$) $\uparrow$ (Mean $\pm$ Std) |
> > > | :--- | :--- | :--- |
> > > | CE Fine-tune (LLM) | 0.881 $\pm$ 0.022 | 0.878 $\pm$ 0.024 |
> > > | RAFT (LLM) | 0.905 $\pm$ 0.019 | 0.902 $\pm$ 0.021 |
> > > | SPR-RAFT (Ours) | **0.935 $\pm$ 0.017** | **0.933 $\pm$ 0.018** |

---

### Decision · Program_Chairs · 2026-04-30

**Decision:**

Accept (regular)

**Comment:**

This paper proposes SPR-RAFT for adapting LLMs to regression tasks via a dual-branch design and hybrid objective. Reviewers agree the problem is meaningful and the method is technically sound, with consistent empirical gains and strong efficiency. While concerns remain about limited novelty, modest margins, and whether LLMs are necessary compared to specialized models, the rebuttal addressed key issues to improve confidence. I recommend weak acceptance.